# Local Extremes of Selected Industry 4.0 Indicators in the European Space—Structure for Autonomous Systems

**Milena Botlíková [1] and Josef Botlík [2,*]** 

[1]  Faculty of Philosophy and Science in Opava, Silesian University in Opava, Bezručovo nám. 14,
    746 01 Opava, Czechia; milena.botlikova@fpf.slu.cz
[2]  School of Business Administration in Karvina, Silesian University in Opava, Univerzitní nám. 1934/3,
    733 40 Karviná, Czechia
[*]  Correspondence: botlik@opf.slu.cz; Tel.: +420-596-398-242

**Abstract:** In the past, the social and economic impacts of industrial revolutions have been clearly identified. The current Fourth Industrial Revolution (Industry 4.0) is characterized by robotization, digitization, and automation. This will transform the production processes, but also the services or financial markets. Specific groups of people and activities may be replaced by new information technologies. Changes represent an extreme risk of economic instability and social change. The authors described available published sources and selected a group of indicators related to Industry 4.0. The indicators were divided into five groups and summarized by negative or positive impact. The indicators were analyzed by precedence analysis. Extremes in the geographical dislocation of factor values were found. Furthermore, spatial dependencies in the distribution of these extremes were found by calculating multiple (long) precedencies. European countries were classified according to individual groups of indicators. The results were compared with the real values of the indicators. The indicated extremes and their distribution will allow to predict changes in the behavior of the population given by changes in the socio-economic environment. The behavior of the population can be described by the behavior of autonomous systems on selected infrastructure. The paper presents research related to the creation of a multiagent model for the prediction of spatial changes in population distribution induced by Industry 4.0.

**Keywords:** Industry 4.0; indicators; precedence analysis

## 1. Introduction

The Fourth Industrial Revolution (Industry 4.0, Work 4.0, etc.) is a designation for the current trend of digitization with related to it automation of production and the labor market changes that it will bring with it. The concept is based on a documentary that was presented at the Hannover fair in 2013. Changes in the labor market are a priority factor, which relates with new controlling, decision-making, and robotic systems.

### 1.1. Theoretical Background

Industry 4.0 is a controversial process by nature and definition given by enabling technologies that allow it to exist as well as the opportunities it brings.

Industry 4.0 has specified (Kagermann et al. 2013) as a process that is currently understood as the fourth industrial revolution based on cyber-physical systems, changing firms' strategies, organization, business models, value and supply chains, processes, products, skills, and stakeholder relationships.

Industry 4.0 will involve the technical integration of cyber-physical systems (CPS) into manufacturing and logistics and the use of the Internet of Things (IoT) and Internet of Services (IoS) in industrial processes. This will have implications for value creation, business models, downstream services and work organization. Kagermann presents the vision, that Industry 4.0 as part of a smart, networked world (Smart Mobility, Smart Logistics, Smart Buildings, Smart Product, and Smart Grids indicated through Smart Factory and CPS). Industry 4.0 is a 'strategic initiative' of the German government that was adopted as part of the High-Tech Strategy 2020 Action Plan.

Based on the findings from the literature review the authors (Hermann et al. 2016) define Industry 4.0 as follows: Industry 4.0 is a collective term for technologies and concepts of value chain organization. Within the modular structured Smart Factories of Industry 4.0 CPS monitor physical processes, create a virtual copy of the physical world and make decentralized decisions. Over the Internet of Things, CPSs communicate and cooperate with each other and humans in real time. Via the Internet of Services, both internal and cross-organizational services are offered and utilized by participants of the value chain.

No conceptual, operative, or universally accepted definition of Industry 4.0 has been identified thus far, because Industry 4.0 is comprised of an estimated more than 1200 enabling technologies, its innovations rapidly become obsolete, it can be applied in a variety of domains, such as smart factories, cities, grids, health applications, homes, spaces, objects, or machines and different disciplines analyzed different subject, (engineering, economics, management, etc.). Moreover, there are various stakeholders (policymakers, managers, entrepreneurs, academics) who have different interests (Büchi et al. 2020).

According to some studies (Liao et al. 2017; Yin Yong and Dongni 2017), the "Industry 4.0" expression ultimately involves adopting industrial automation systems that assist in managing the value and supply chains, and more widely manage all their related processes, it is possible to determine certain common elements, (application of automation systems, connections between the physical and virtual worlds, the recognizing of a set of enabling technologies, digitalization, the Internet, changes in the relationships with stakeholders, and in governance), these will assist in determining a definition to better encompass the phenomenon.

This trend is often debated under the label Industry 4.0. A key claim put forward in these debates is that Industry 4.0 represents a revolution that will reshape manufacturing industries akin to previous industrial revolutions. This clarification of the identity of Industry 4.0 adds to a better understanding of the relationship between manufacturing and politics as well as technological change in manufacturing (Reischauer 2018).

Industry 4.0 and his implementation indicate changes in business paradigms and manufacturing models, which will be reflected on all levels of manufacturing processes, as well as supply chains, including all workers in the manufacturing process, managers, cyber-physical system designers, and end-users. The implementation strategy of Industry 4.0 means introducing self-automation, self-configuration, self-diagnosis and problem-solving, knowledge, and intelligent decision-making.

The conclusions point to changing business paradigms, legal issues, resource planning, security issues, standardization issues, etc. Implementation Industry 4.0 includes all participants in the production chain, from the manufacturer to the end-users (Karabegović et al. 2020).

### 1.2. Industry 4.0, Unemployment, Labor Market

From the point of view of economic theory, unemployment caused by 'obsolescence' of jobs is structural unemployment; thus, it is natural part of unemployment. It cannot be avoided that as a result of technological progress, some jobs will lose their relevance and will be replacing human power by technology, as in previous industrial revolutions, threatens employment. According to the Ministry of Labor and Trade of the Czech Republic, routine, manual, and physically demanding work will be replaced by technology (warehouse workers, cashiers). Some professions can also be expected to disappear. At the same time, however, new positions should be created that will require a continuous "acceleration of technological adaptability" (CT24 2018).

Compared to previous industrial revolutions can be assumed inequality of impacts on employment, it will be reflected in job losses across working position, both in unskilled work and in highly specialized activities (diagnostic medical systems, legal systems). Significant problems can be expected in the services segment, replacing human power with information kiosks, coffee and food vending machines, Internet search services, call centers, writing and sending orders, etc. There is also a definite shift in the level of trade, the emergence of self-service cash registers, Scan & Go systems, interconnection of physical stores with online e-shops, creation of a network of ticket offices and showrooms (CT24 2018). Significant changes can also be expected in warehousing and logistics, for example fully automated warehouses, that will require a minimum of human labor. Moving between working positions can be expected, which is conditional regarding lack of qualifications, into the segment outsourcing, Software as a Service, etc. According to Deutsche Bank, the Fourth Industrial Revolution will also cover so-called white collar and financial innovations related to the development of cryptocurrencies (Tůma 2017). Changes in the geopolitical and social distribution of the population induced by the Fourth Industrial Revolution can be predicted to some extent. According to (prumysl-4.cz, Industry 4.0 2019), the productivity of production will increase by up to 30% after technological changes, up to 40% of the population will have to change their qualifications.

However, given the diversity and complexity of socio-economic links, the prediction is more complicated, with a considerable degree of uncertainty. The changes will take place continuously in several stages. In the first stage, jobs should decline. There will then be an increase in the demand for highly qualified workers to take care of the operation of the machines. There is this place for the creation of jobs in information technology (IT), development and marketing communication. Foreign experience shows that 2.5 new jobs should be added per lost job. Low-skilled jobs are likely to be lost, for example in assembly line production (Korbel 2015).

A comprehensive survey of the job vacancies induced by Industry 4.0 was conducted by some authors (Pejic-Bach et al. 2020). Analysis of the job advertisements revealed that most of them were for full time entry; associate and mid-senior level management positions and mainly came from the United States and Germany. Text mining analysis resulted in two groups of job profiles. The first group of job profiles was focused solely on the knowledge related to Industry 4.0: cyberphysical systems and the Internet of Things for robotized production; and smart production design and production control. The second group of job profiles was focused on more general knowledge areas, which are adapted to Industry 4.0: supply change management, customer satisfaction, and enterprise software.

It is obvious that the projected impacts vary and evolve over time, while in 2015 the highest potential was seen in the IT sector, in 2016 attention shifted to the service segment (Čičváková 2017) and in 2017 to the financial and trade sectors (Tůma 2017). One of the possible tools for prediction is simulation using autonomous systems that simulate the behavior of real populations. According to (CT24 2017a), if we predict the development of jobs, 400,000 jobs will be lost in the Czech Republic, and another 1.5 million will undergo fundamental changes. According to (CT24 2017b), it is predicted that up to 53% of jobs in the Czech Republic will be lost. According to (Čičváková 2017), this figure can be called into question as the changes will be gradual and new jobs will be created. Above all, these will be IT positions. According to Linked (Linkedin 2019), there will be increased demand for professions in cloud and distributed computing, statistical analysis and data mining, web architecture and development framework, middleware and integration software, user interface design, network and information security, mobile development, data presentation, SEO/SEM marketing, or storage systems and management.

Already today it is possible to trace the requirements in the structure of the production process in areas such as Internet of Things, artificial intelligence, cloud applications, big data, unified data storage, system engineering, drawingless production (digital models), reverse engineering, additive production, 3D printing, etc., which are closely related to simulation processes and systems engineering. The professional public disagrees on the degree of dominance of IT technologies and the degree of 'robotics'. The identification of vulnerable groups is problematic because it is not possible to accurately

estimate the rate of labor market absorption resulting from the supply of new jobs. However, it is clear that these will be increasingly IT-related positions. Partially, it is possible to quantify and identify endangered job positions; these will primarily include positions in the services and robotizable production segment.

Job position and labor market may also be subject to increased security risks. It is possible to predict critical industrial activities within Industry 4.0 and potential adverse impacts on business performance due to breaches of cybersecurity. In particular, cybersecurity is endangered in terms of loss of confidentiality, integrity, and availability of data associated with networked manufacturing machines (Corallo et al. 2020). Many experts anticipate significant impacts in the supply chain management segment (Dev et al. 2020; Hofmann et al. 2019), which can be identified as security risks associated with the interconnection cyber processes into international digital structures.

*1.3. Literature Review and Current State of Knowledge*

Some authors (Liao et al. 2017) have made systematically review the state of the art of this new industrial revolution wave. The aim of their study is to address this gap by investigating the academic progresses in Industry 4.0. A systematic literature review was carried out to analyze the academic articles within the Industry 4.0 topic that were published online until the end of June 2016. These results summarize the current research activities (e.g., main research directions, applied standards, employed software, and hardware) and indicate existing deficiencies and potential research directions. among the 10 most frequent words by the authors are system(s), production(s), manufacturing, industrial, technology(ies), smart, physical, factory(ies), data, and concept(s).

The analysis of about 35 documents from 2008–2018 was carried out by (Machado et al. 2019) with the aim to a systematic review of the links between sustainable manufacturing research and the Industry 4.0 conceptual framework. The results are generalized to the conclusion that current research is in line with the objectives set by the various national industrial programs. The authors point out that the absence of national and regional programs significantly reduces the opportunities offered. In this connection, it should be noted that the Ministry of Industry and Trade elaborated on the Czech Republic Initiative of Industry 4.0 (Ministry of Industry and Trade 2016) with the aim sustaining and strengthening the competitiveness of the Czech Republic throughout the Fourth Industrial Revolution. The initiative was approved by the Czech government on 24 August 2016. Scientific publications examine the effects and impacts of Industry 4.0 from different angles (Prinz et al. 2018).

Due to the worldwide spread of digital technologies in the manufacturing industry, some authors (Reischauer 2018; Valenčík 2019) consider Industry 4.0 only an innovative process, a policy-driven innovation discourse in manufacturing industry that aims to institutionalize innovation systems that encompass business, academia, and politics. There are claims too, that Industry 4.0 is listed only as the first of eight core fields within this area ahead of smart services, smart data, and cloud computing (Reischauer 2018), and the authors point out, that most important proponents in the debate on Industry 4.0 are the German federal government and German federal ministries.

There is considerable interest in the so-called Fourth Industrial Revolution, but this concept is not clear in the literature. Literary research on Industry 4.0 was also carried out by, for example, (Fonseca 2018), which aims to present an overview of several industrial revolutions with an emphasis on Industry 4.0. The authors conclude that Industry 4.0 can help organizations reach new and emerging markets through a differentiation strategy or even create new business models. However, for most companies it is still in its early stages, and digital transformation will require strong leadership, the right human skills and overcoming specific barriers to be successfully implemented. Although according to the authors, this will lead to a significant improvement in job creation, there will also be significant job losses for low-skill employees. Whereas, according to the authors: by 2018, 40% of companies in the European Union still had not adopted any of the new advanced digital technologies, it is necessary to further explore the factors that can accelerate new trends.

Some authors (Büchi et al. 2020) have been identifying the main characteristics of the pillars of Industry 4.0 enabling technologies, and particularly their definitions and the opportunities they offer, operationalizing the concepts of openness and performance and empirically verifying the causal relationship between the degree of openness to Industry 4.0 and performance. The authors did a literature review of 249 articles and identified the origins and definitions of Industry 4.0, as well as the key factors and opportunities related to the pillars of its enabling technologies.

The analysis is conducted using a sample of 231 Italy manufacturing industry units developing the Industry 4.0 concept in Piedmont (northern Italy) in 2018.

Some of the authors analyze possible concrete impacts in terms of technology. For example, Zhang et al. (2019) see the main impacts especially in the production process when targeted robotization occurs. Businesses will be forced to transform traditional production into intelligent factories with computer systems to create physical products. Zhang et al. (2019) presents the architecture of using the ubiquitous cloud-based robotic systems to intelligently produce a customized product. The authors point that the proposed approach can achieve the goal of intelligent production and customized product development. Some authors point out the impacts in terms of the labor market and employment. They point to contradictions where digitization and automation for selected activities can either eliminate the competitive environment, thereby contributing to an increase in the quality of life (increased wages, well-being) or create a threat of unemployment which, together with low incomes, will have negative effects on wellbeing and mental health.

Other authors point out the lack of a comprehensive overview and highlight associated risks, especially on a framework of risks in the context of Industry 4.0 that is related to the Triple Bottom Line of sustainability. Risks mentioned include economic risks (Birkel et al. 2019), the risks associated with high or false investments are outlined, the threatened business models, and increased competition from new market entrants. Additionally, risks can be associated with technical risks, e.g., technical integration, information technology related risks such as data security, and legal and political risks, such as unsolved legal clarity in terms of data possession.

A comparative study (Büchi et al. 2020) helps predict the use of technology-based services, and tests the predicting capacity of demographic factors, attitudes toward technology, and cultural values. The results indicate that demographic variables have the highest predictive capacity. Attitudes toward technology demonstrate some predictive ability, while cultural values have a negligible direct impact on technology use. The results of structural equation models indicate that cultural values have a fundamental indirect impact on the use of technology-based services.

In many cases, there are studies where the stress and threat of job loss caused by automation and digitization induce negative behavior in the workplace, such as (Coldwell 2019). The paper is based on secondary data analysis; the theoretical model suggests that extreme forms of behavior associated with the digital era can create organizational entropy.

Furthermore, the loss of social interaction as tasks are given increasingly to computers and automated services is shown as a further social risk (Birkel et al. 2019). Some employees might struggle to spend more time in front of the computer, and less time interacting with other humans.

Risk factors related to management behavior, specifically responsibility and ethics, are mentioned and reported by (Kliestik et al. 2018). Other approaches to this issue are applied, for example, by (Yun and Zheng 2019), focusing on understanding the open innovative micro and macro dynamics for social, environmental, economic and cultural policies and knowledge sustainability. It also provides an overview of the sustainability of the economy, society, and the environment during the Fourth Industrial Revolution. The author gains the initial knowledge mainly of literature search. Another specific area examined in the context of Industry 4.0 is the significant social and economic opportunities and challenges that require governments to respond appropriately to support the transformation of society. The aim of the study, which (Manda and Soumaya 2019) conducted, is identify challenges that developing countries face in adopting digital transformation programs to take advantage of the social and economic benefits of the industrial revolution 4.0. The research is based on an interpretative case

study that uses data, third-party documents and literature review as the primary data collection method. The case study focuses on South Africa, the only developing country to adopt digital transformation as its primary growth strategy.

European experience with shaping and regulating socially responsible behavior of economic subjects; Galetska et al. (2019) looked at differentiating the dominant driving forces of corporate social responsibility and strategic priorities in Germany in the Industry 4.0 environment. They identified interconnected types of responsibility (legal, economic, professional, moral, political, etc.) that reflected the system of company values. They draw attention to the deterioration of competition that worsens conditions of economy. They point out the need for modeling based on a strategy of sustainable development, socially responsible behavior of economic structures. The authors also identified the problem of the absence of mechanisms for maintaining social compromise in society. The system of ensuring the responsibility of the company's entities for shaping the normal living conditions of the company is one of the institutional mechanisms of social control and creates conditions for balancing personal, collective, and social interests.

Other studies (Masud et al. 2019) have found that economic responsibility has no intervening role while environmental and social responsibility significantly mediated the relationship between organizational strategic performance and corporate social responsibility performance.

Institutional support for the functioning of mechanisms to promote social compromise with regard to the creation of normal living conditions is based on the levers of state regulation (subsidies, preferential taxation, economic incentives, and compliance with standards of activity), the institutions of business (international and national standards for business), and public interest (social systems, spatial planning). The conclusions of this work clearly show the potential overlap of the impact of Industry 4.0 on trans-regional ties and the need to investigate the impact of Industry 4.0 also in the European or transnational context, as the consequences may cause social dissatisfaction leading to economic instability of the region.

Similar conclusions are provided by Veselica (2019), he carries out the description, compares and analyzes the indicators of competitiveness and innovation for the selected national economy—the Republic of Croatia. The aim is to identify regional competitiveness based on the EU Regional Competitiveness Index indicator for 2016. The author identifies the relationship between innovation and competitiveness for the national economy in the European context.

Severe globalization features and impacts have been identified in particular by (Efremov and Vladimirova 2019). Especially in IT, education a very important factor in the processes related to Industry 4.0. Helmi et al. (2019) draw attention to the need for transformation in the preparation of experts with contextual knowledge (STEM—a concept focused on four disciplines—Science, Technology, Engineering, Mathematics), and preference for higher order thinking skills (HOTS—higher order thinking skills is a concept of education reform based on learning taxonomies). Comparative methods are used, for example, by Storolli et al. (2019), which performs a comparative analysis of technology tools between Industry 4.0 and Smart City. It also focuses on environment and sustainable planning. Based on the knowledge Smart City deduces the concept of strategic development conditional on the introduction of IT.

Comparisson as a method is also used, for example, with Min et al. (2019), their research points out the need for transnational and global monitoring of changes induced by the Fourth Industrial Revolution, and draws attention to the necessity of defining the sites. It provides a quantitative comparison of developed countries, strategies of Germany, USA, China, Japan, and Korea. The purpose of this paper is to analyze the side effect of information and communication technology (ICT) implementation. This study attempted to provide a theoretical approach to identifying national policy strategies and to show the practical implications and impacts of the Fourth Industrial Revolution by comparing the effects of ICT industries.

The national policies need is also highlighted by other authors (Büchi et al. 2020), Table 1 shows selected national concepts.

**Table 1.** Summary—National concepts, (Büchi et al. 2020), modified.

| Country | Industrial Plan |
| --- | --- |
| Germany | High-Tech Strategy 2020 |
| France | La Nouvelle France Industrielle (The New Industrial France) |
| United Kingdom | Future of Manufacturing |
| United States | Advances Manufacturing Partnership |
| Czechia | Industry Initiative 4.0 |
| Slovakia | Proposal of the Intelligent Industry Action Plan of the Slovak Republic |
| China | Made in China 2025 |
| Singapore | Research, Innovation and Enterprise |
| South Korea | Innovation in Manufacturing 3.0 |
| Italy | Impresa 4.0 |

Some studies (Efremov and Vladimirova 2019) use multiple methods of scientific knowledge: analysis, synthesis, generalization (to reveal the conceptual-categorical apparatus of the research subject); statistical methods, grouping, empirical approach (when analyzing the practice of allocating social responsibility among social partners in the EU to ensure social protection of the population and differences between EU countries, etc. The research methodology is based on defining the general principles of construction of the corporate and states social responsibility system, revealing the nature and apparatus of Industry 4.0, taking into account its main theoretical concepts.

To mention Czech resources, it is necessary to mention (Koren 2018) who point out that changes in the nature of work, its organization and forms naturally affect the performance of specific types of qualifications and competences. This leads to changes in labor market demand and, in relation to its potential, to changes in the placement and cost of human labor. Social impacts can be monitored in the areas of employment, legislation, tax policy, and the content and form of education. It is therefore clear that these authors also point out possible spatial changes in the labor market. From Czech sources, we can also mention a monograph (Valenčík 2019), in which the authors question the importance of Industry 4.0, as the main reasons stated that the concept of the so-called Fourth Industrial Revolution is a manifestation of inertia thinking at a turning point of time. He further claims that the concept is shallow and uncomplex. It only observes some external and irrelevant phenomena, it does not try to grasp all the essential aspects of the present time, nor is it equipped to do so. According to the authors, the concept of the Fourth Industrial Revolution is being misused to confuse, to real camouflage problems and their causes, to ideologically sterilize political forces and political entities, and to increase the disorientation of people.

Table 2 summarizes selected cited sources. The focus on Industry 4.0 is obvious; the share of quoted sources from Q1 and Q2 journals is high, complemented by regional sources and indexed conferences and professional publications. The table shows the positives and negatives of resources, key areas and scientific methods. (WoS Con.—WoS Conference, J Q1, Q2—Journal Q1, Q2, Prof. P.—professional periodical, Prof. I. P.—professional internet periodical, Mon.—scientific monographic, SD—strategic documents).

**Table 2.** Summary—literature sources.

| Study | Circuit | Methods | Benefits of the Study | Negative of the Study | Specification of Factors |
|---|---|---|---|---|---|
| Bai (2013) WoS Con. | Spatial mobility, circulation movement, tourism | Multi-agent systems, simulation | Problem identification: system analysis is no longer enough to describe the spatial architecture of the industry | addressed in the conceptual framework for the intelligent tourism information system | Environmental, sociocultural, and economic impacts, |
| Birkel et al. (2019), Jour. Q3, Q2 | Industry 4.0 Triple Bottom Line, risk management | | Risks identification that arise within the framework of Industry 4.0 | empirical study | Ecological, environmental |
| Braccini and Margherita (2019), Jour. Q3, Q2 | Industry 4.0 Triple Bottom Line | Case study | Confirmed the relevance of the factors affecting the individual | Only the case study | Ecological, environmental |
| Büchi et al. (2020), Jour.Q1 | Industry 4.0 | Regression models | Practical research | Only Italy comparation | Used technologies, value chain, future investments, perceived opportunities |
| Coldwell (2019) Jour. Q1, Q2 | Digitization, automation | Secondary data analysis | Theoretical model, extensive literature research | general conclusions | Employment, social environment, labor market, corporate social responsibility, |
| Corallo et al. (2020) Jour Q1 | Business performance, cybersecurity | Analysis, correlation | Knowing and evaluating in advance the main critical assets | General | Employment, social environment, manufacturing, digitalization |
| Cruz-Cárdenas et al. (2019) Jour.Q1 | Technology, demographics, cultural value economy | Comparison structural equation models | Comparative study | Only two states Ecuador and Russia | Demographic factors, technology, cultural |
| Čičváková (2017) | Industry 4.0, terminology, terms | Description | Summary of terms | General | Labor market |
| Deng et al. (2018) Jour. Q1, Q2 | Allocation of resources | Multi-agent systems, | Modeling with agents | Not directly related to Industry 4.0 | – |
| Dev et al. (2020) Jour. Q1 | Industry 4.0, reverse logistics, statistic methods | Mathematical modeling | Instructions for managements | Hypothetical case | Environmental and economic |
| Efremov and Vladimirova (2019) WoS. Con. | Globalization, globalization factors, | Analysis, compression | Analysis of globalization benefits | General conclusions | Globalization index |
| Fonseca (2018) WoS Con | Industry 4.0, impacts of digitization | Literature research, identification of keywords | Summarizes political, economic, social, technological, environmental and legal issues, concretization of strategies and new business models | General conclusions | Spectrum of Industry 4.0-related factors, |
| Galetska et al. (2019) Jour. WoS | Industry 4.0, globalization, social responsibility, sustainable development | Compaction, summation, time series | Specification of social responsibility | Low data uptime | Social environment, social responsibility, degree of globalization, employers' social contribution |
| Helmi et al. (2019) WoS Con | Industry 4.0, education | Systematic describes | Strengthening learning in STEM education | Narrow population group | Educational factors |
| Hermann et al. (2016) WoS Con | Industry 4.0, reverse logistics, | Systematic literature review | Comprehensive literature search | General conclusions | Smart Factory, IoS, IoT |
| Hofmann et al. (2019) | Supply chain management 4.0 Industry 4.0 | literature review, summary of supply chain management | specifically designed topics for academic research | general facts | Customer factors, |
| Charnley et al. (2019) Jour. Q3, Q2 | Simulations, circulatory processes | Discrete simulation, primary analysis | Circular economic model focusing on product life | Close focus (Great Britannia, automotive industry) | Economic, manufacturing, digital intelligence |
| Kliestik et al. (2018) Jour. Q1, Q2 | Finance, banking, bankruptcy | Robust analysis, prediction tools | Prediction model, specification of risk factors | Close focus on banking | Factors related to management behavior, risk factors |
| Korbel (2015) Prof. P. | Industry 4.0, genus of production | Description | Information character | Informative character | – |

**Table 2.** *Cont.*

| Study | Circuit | Methods | Benefits of the Study | Negative of the Study | Specification of Factors |
|---|---|---|---|---|---|
| Koren (2018) WoS Con | Automation, manufacturing, services | Statistical data analysis | Data analysis of the labor market in the Czech Republic | Narrow focus, mostly | Social impacts, employment, legislation, education |
| Kagermann et al. (2013), SD | Industry 4.0 | Theory, description | Strategic materials | Germany strategic | Industry, innovation |
| Karabegović et al. (2020), WoS Con | Business paradigms, Industry 4.0 | Theory description | Industry 4.0 business data analysis | general specifications | Manufacturing process |
| Liang et al. (2017) WoS Con | Economic growth | Simulation, analytical pathway | Prediction of economic growth | general conclusions | Economic growth factors |
| Liao et al. (2017), Jour. Q1, Q2 | Industry 4.0 | Systematic literature review | Identification of Industry 4.0 key expressions | only the comparison | Basic data, keywords |
| Machado et al. (2019) Jour. Q1, Q2 | Sustainable manufacturing, Industry 4.0 | Literature review | Identification of Industry 4.0 key expressions | only the comparison | Group of technological factors |
| Masud et al. (2019), Jour. Q1, Q2 | Triple Bottom Line, organizational strategy | Structured questionnaire, literature review | Business management, especially in the policy and strategy area | Only a sample of 250 employees from Bangladesh | Social responsibility, strategic performance |
| Manda and Soumaya (2019) WoS Con | Industry 4.0, developing countries, state concessions | Comparison, description, analysis | Analysis of the national strategy | focus on South Africa | Socio-technical |
| Min et al. (2019) Jour. Q1 | Industry 4.0, innovation | Comparative study | Specification of practical implications | theoretical approach to the determination of national policy strategies is not complete | Factors related to ICT |
| Pejic-Bach et al. (2020), Jour. Q1 | Industry 4.0, employment | Topic mining | Comprehensive survey of demanded jobs | only text mining without feedback | Human resource management, education, smart factory |
| Prinz et al. (2018) Jour. Q1, Q2 | Industry 4.0, Smart Factory | Comparison | Summary information about Industry 4.0 | general specifications | – |
| Reischauer (2018), Jour. Q1 | Identity of Industry 4.0 long wave theory | Comparison, description | Interdisciplinary work | Very old citation | Triple helix factors, technology and innovation factors |
| Storolli et al. (2019) WoS Con | Industry 4.0, Smart City | Comparison, description | Smart City specifications | general specifications | Factors and keywords related to Smart City |
| Tang and Yi (2018) Jour. Q1, Q2 | Multiagent approach, coordination, distributed optimization problem | Simulation | Use of local information for analyses | limited interpretation circuit | – |
| Tůma (2017) Prof. I. P. | Industry 4.0, positives, negatives | Description | Information character | informative character | General recommendations |
| Valbuena et al. (2008) Jour. Q1, Q2 | Agriculture, multi-agent analyses | Case study, multiagent systems | Multi-agent systems model | Method described in this paper has some limitations | Production scale, environmental, social (lifestyle) |
| Valenčík (2019) Mon. | Comparative factual characteristics of the 21st century | Political economic analysis | Industry 4.0 Criticism | Lack of practical conclusions | Comprehensive set of recommended identifiers |
| Veselica (2019) WoS Con | Industry 4.0, competitiveness | Complications, analysis | comprehensive comparison | Limited interpretation | Indicators for competitiveness and innovation, Global Competitiveness Index |
| Yang et al. (2019) Jour. Q1 | Energy | Distributed optimization of multi-agent systems | Detailed overview of existing distributed optimization algorithms, coordination of distributed energy resources | Close focus, general conclusions | Energy economic factors |
| Yun and Zheng (2019) Jour. Q3, Q2 | Industry 4.0, sustainability, innovation ecosystem | Literature review and analysis | Comparisons of industry, education, government, and society | Conceptual model needs to be further validated | Social, environmental, economic, cultural, policy, and knowledge sustainability, Innovation |
| Zhang et al. (2019) Jour. Q1 | Industry 4.0, customer, marketplace, Cloud systems | Literature review, framework analysis, case studies | Real experiment | Experiment built on simplified reality | ICT factors, industry factors, marketplace indicators |

*1.4. Research Question and the Proposed Procedure*

It follows from the foregoing that changes in the labor market and in the social field will depend on the level of education and ability to realize the potential of IT. The question is whether the impact of Industry 4.0 on society can be predicted based on changes in selected factors. It has been hypothesized that it is possible to prove a correlation between selected factors related to Industry 4.0. The research focused on factors related mainly to unemployment in the regions and migration of the workforce related to IT and digitization. The research is primarily focused on EU countries.

Based on experience from other studies, a multiagent analysis based on the examination of changes using autonomous systems was chosen as the method.

Multiagent analysis is based on computer modeling and simulation. The outcome of this analysis are multiagent models, these models fall into more general category of multiagent systems, they are mainly used to simulate complex systems in various fields of interest (economics, biology, social sciences), which are difficult to grasp by other research methods. The principle of multiagent simulation is based on the use of agents, which are software autonomous entities with relatively simple behavior.

This method is based, for example, on research by (Tang and Yi 2018), who present a passivity-based analysis using only the local target function, tracking local data and exchanging information from their neighbors. (Liang et al. 2017) proved suitability of application of multiagent systems to economic processes, the aim was to create a model for analysis of contemporary economic structures. In terms analysis of migration can be found connection with simulations of the process of architecture and development of tourism with help multiagent systems (Bai 2013). Other suitable examples of using multi-agent systems can be found in (Yang et al. 2019), (Deng et al. 2018), (Valbuena et al. 2008), and (Charnley et al. 2019).

Due to the high demands on data processing, two basic reductions were chosen. The actual research monitors increase and decreases of factor values, thus enabling data processing by binary matrices. Factor values are compared between states that are either contiguous (they have a shared border) or geographically close. Agents move between related states looking for increases and decreases in the values of selected factors in order to identify the longest paths. The following partial steps were chosen for the research itself. In the first phase, an infrastructure for agent movement on a set of selected states is generated. Subsequently, year-on-year changes are identified for individual factors, which are compared on the generated infrastructure. For each country, precedents are recorded for each factor and the number of precedents for each country is determined. It indicates how many states in the neighborhood have a lower value of the given factor. Furthermore, precedents of multiple length are identified, which identify pairs of non-neighboring states with maximum and minimum values of the given factor, among which there is a constant increase in the values of the monitored variable (factor). Factors are represented by vectors, the precedence using square matrices and the multiple precedences using matrices of matrices. Finally, the number of precedencies in individual countries for individual factors is compared.

## 2. Data Sources and Methods

Precedence analysis was chosen as an analytical method. The precedence analysis is based on an analysis of the increase in the values of the monitored factors among subjects with a defined binding. Neighboring states or relative proximity are used as linkages in this analysis. Immediate increases (first alphabet) of factor values are indicated. Further, multiple (long) precedencies are indicated, which consist of continuous, consecutive, first precedencies. The numbers of the first precedencies indicate the significance of the given factor in the immediate vicinity of the selected state. Long precedencies indicate areas where there is a multiple, sustained increase in the value of a given factor in area. The numbers of long precedencies and the existence of the longest precedencies indicate the dominance of the country under review in a given factor in a wider geographical context (indicating an increase in the values of the monitored factor in area). Precedence analysis complements classical comparative analyses with a spatial context because it takes into account the geographical distribution of states.

For the proposed method of analysis (precedence analysis) it was necessary, based on the performed literature search, to provide meaningful and verified data. Due to the type of analyses, it was not recommended to perform primary data collection, data was extracted from existing professional databases. At the same time, it was necessary to reduce the set of factors identified in the literature search used by the cited authors.

*2.1. Basic Data*

Although the authors know that one of important factors of Industry 4.0 is sustainability (Masud et al. 2019; Birkel et al. 2019), especially in the context of all dimensions of the triple bottom line (TBL)—ecological, social, and economic, environmental factors were not investigated because the required data series were not available. Environmental dimension of the TBL was describe of another authors (Braccini and Margherita 2019), these authors deal with the sustainable use of resources. On the contrary, factors related to education, science and research were examined. A group of 29 factors related to Industry 4.0 were selected for this contribution. The data were examined for the years 2010–2018, 261 data series were analyzed in total. The Eurostat database was used as a data base. The factors are listed in Table 3.

**Table 3.** Factors used for the analysis.

| Index | Factor | Effect | Note | Group |
|:---:|:---:|:---:|:---:|:---:|
| 1 | Total employment (resident population concept—LFS) | + | Percentage of total population, age group 20–64, total | Em |
| 2 | Total employment (resident population concept—LFS) | − | Percentage of total population, age group 20–64, female | Em |
| 3 | Gross domestic expenditure on R&D (GERD) | + | Percentage of gross domestic product (GDP) | RD |
| 4 | Early leavers from education and training by sex | − | From 18 to 24 years, total | Edu |
| 5 | Early leavers from education and training by sex | + | From 18 to 24 years, female | Edu |
| 6 | Tertiary educational attainment | + | From 30 to 34 years, total | Edu |
| 7 | Tertiary educational attainment | − | From 30 to 34 years, female | Edu |
| 8 | Resource productivity | + | Euro per kilogram, chain linked volumes (2010) | Eco |
| 9 | Purchasing power standard (PPS) per kilogram | − | | Eco |
| 10 | Index resource productivity | + | Index, 2000 = 100 | Eco |
| 11 | Eco-innovation index | + | Index, EU = 100 | RD |
| 12 | People at risk of poverty or social exclusion | − | Percentage of total population, | Soc |
| 13 | People at risk of poverty after social transfer | − | At risk of poverty rate (cut-off point: 60% of median equivalized income after social transfers) | Soc |
| 14 | Severely materially deprived people | − | Percentage | Soc |
| 15 | Agriculture, forestry and fishing | − | Percentage of total (based on persons) | Eco |
| 16 | Industry (except construction) | − | Percentage of total (based on persons) | Eco |
| 17 | Construction | − | Percentage of total (based on persons) | Eco |
| 18 | Wholesale and retail trade, transport, accommodation, and food service activities | − | Percentage of total (based on persons) | Eco |
| 19 | Information and communication | + | Percentage of total (based on persons) | Eco |
| 20 | Financial and insurance activities | − | Percentage of total (based on persons) | Eco |
| 21 | Real estate activities | − | Percentage of total (based on persons) | Eco |
| 22 | Professional, scientific, and technical activities; administrative and support service activities | | Percentage of total (based on persons) | Eco |
| 23 | Public administration, defence, education, human health and social work activities | − | Percentage of total (based on persons) | Eco |
| 24 | Arts, entertainment and recreation; other service activities; | − | Percentage of total (based on persons) | Eco |
| 25 | HRST: Persons with tertiary education (ISCED) and/or employed in science and technology | + | Percentage of active population, From 15 to 74 years | RD |
| 26 | SE: Scientists and engineers | + | Percentage of active population, From 15 to 74 years | RD |
| 27 | HRSTO: Persons employed in science and technology | + | Percentage of active population, From 15 to 74 years | RD |
| 28 | HRSTE: Persons with tertiary education (ISCED) | + | Percentage of active population, From 15 to 74 years | RD |
| 29 | HRSTC: Persons with tertiary education (ISCED) and employed in science and technology | + | Percentage of active population, From 15 to 74 years | RD |

Industry 4.0 has the potential for an enormous change in the entire value creation, which would be both positive and negative (Birkel et al. 2019). The effect indicates a positive (+) or negative (−) effect on the population. The group indicates what type of indicator it is (edu—education, em—employment,

eco—economic factors, rd—science and development, soc—social). Factors 15–24 related by "Total employment domestic concept", are therefore included in the social group, because they affect the distribution of the population in individual professions and thus can affect the social composition of the population. The negative influence of factors 2, 7, etc. is given by the fact that women generally have a less favorable relationship with ICT and robotics.

*2.2. Factors by Eurostat*

Factors 1 and 2 by Eurostat definition: The employment rate is calculated by dividing the number of persons aged 20–64 in employment by the total population of the same age group. The indicator is based on the EU Labor Force Survey. The survey covers the entire population living in private households and excludes those in collective households such as boarding houses, halls of residence, and hospitals. Employed population consists of those persons who during the reference week did any work for pay or profit for at least one hour, or were not working but had jobs from which they were temporarily absent. Factor 2 summarizes only women.

Factor 3 by Eurostat definition: The indicator provided is GERD (gross domestic expenditure on R&D) as a percentage of GDP. "Research and experimental development (R&D) comprise creative work undertaken on a systematic basis in order to increase the stock of knowledge, including knowledge of man, culture and society and the use of this stock of knowledge to devise new applications".

Factors 4 and 5 by Eurostat definition: Early leaver from education and training, previously named early school leaver, refers to a person aged 18 to 24 who has completed at most lower secondary education and is not involved in further education or training; the indicator 'early leavers from education and training' is expressed as a percentage of the people aged 18 to 24 with such criteria out of the total population aged 18 to 24.

For Eurostat statistical purposes, an early leaver from education and training is operationally defined as a person aged 18 to 24 recorded in the labor force survey (LFS):

- Whose highest level of education or training attained is at lower secondary education. At most lower secondary education refers to ISCED (International Standard Classification of Education) 2011 level 0–2 for data from 2014 onwards and to ISCED 1997 level 0–3C short for data up to 2013;
- Who received no education or training (neither formal nor non-formal) in the four weeks preceding the survey.

The 'early leavers from education and training' statistical indicator is then calculated by dividing the number of early leavers from education and training, as defined above, by the total population of the same age group in the Labor force survey. Factor 5 summarizes only women.

Factors 6 and 7 by Eurostat definition: The indicator is defined as the percentage of the population aged 30–34 who have successfully completed tertiary studies (e.g., university, higher technical institution, etc.). This educational attainment refers to ISCED (International Standard Classification of Education) 2011 level 5–8 for data from 2014 onwards and to ISCED 1997 level 5–6 for data up to 2013. The indicator is based on the EU Labor Force Survey. Factor 7 summarizes only women.

Factor 8 by Eurostat definition: The indicator is defined as the gross domestic product (GDP) divided by domestic material consumption (DMC). DMC measures the total amount of materials directly used by an economy. It is defined as the annual quantity of raw materials extracted from the domestic territory of the local economy, plus all physical imports minus all physical exports. It is important to note that the term 'consumption', as used in DMC, denotes apparent consumption and not final consumption. DMC does not include upstream flows related to imports and exports of raw materials and products originating outside of the local economy.

The indicator is part of the Resource Efficiency Scoreboard. It is used to monitor progress towards a resource efficient Europe. Resource productivity is the lead indicator of the scoreboard.

Factor 9 by Eurostat definition: The purchasing power standard, abbreviated as PPS, is an artificial currency unit. Theoretically, one PPS can buy the same amount of goods and services in each country.

However, price differences across borders mean that different amounts of national currency units are needed for the same goods and services depending on the country. PPS are derived by dividing any economic aggregate of a country in national currency by its respective purchasing power parities. PPS is the technical term used by Eurostat for the common currency in which national accounts aggregates are expressed when adjusted for price level differences using PPPs. Thus, PPPs can be interpreted as the exchange rate of the PPS against the euro. Standard purchasing power (PPS) capable of expressing weak economic aggregates in international comparisons. It makes a simple distinction between how many currency units (PPS) can be obtained through the quantity of goods and services in each country.

Factor 10 by Eurostat definition: The indicator is defined as the gross domestic product (GDP) divided by domestic material consumption (DMC). It is used to monitor progress towards a resource efficient Europe. Resource productivity is the lead indicator of the Scoreboard DMC measures the total amount of materials directly used by an economy. It is defined as the annual quantity of raw materials extracted from the domestic territory of the local economy, plus all physical imports minus all physical exports. It is important to note that the term 'consumption', as used in DMC, denotes apparent consumption and not final consumption. DMC does not include upstream flows related to imports and exports of raw materials and products originating outside of the local economy.

Factor 11 by Eurostat definition: The indicator is based on 16 sub-indicators from eight contributors in five thematic areas: eco-innovation inputs, eco-innovation activities, eco-innovation outputs, resource efficiency outcomes, and socio-economic outcomes. The overall score of an EU Member State is calculated by the unweighted mean of the 16 sub-indicators. It shows how well individual Member States perform in eco-innovation compared to the EU average, which is equated with 100 (index EU = 100).

The index complements other measurement approaches of innovativeness of EU countries and aims to promote a holistic view on economic, environmental, and social performance. The relevant target in the Roadmap to a Resource Efficient Europe is for an increase in the funding for research that contributes to the environmental knowledge base.

Factor 12 by Eurostat definition: This indicator corresponds to the sum of persons who are: at risk of poverty or severely materially deprived or living in households with very low work intensity; only counted once even if they are present in several sub-indicators. At risk-of-poverty are persons with an equivalized disposable income below the risk-of-poverty threshold, which is set at 60% of the national median equivalized disposable income (after social transfers). Material deprivation covers indicators relating to economic strain and durables. Severely materially deprived persons have living conditions severely constrained by a lack of resources, they experience at least 4 out of 9 following deprivations items—People living in households with very low work intensity are those aged 0–59 living in households where the adults (aged 18–59) work 20% or less of their total work potential during the past year.

Factor 13 by Eurostat definition: The persons with an equivalized disposable income below the risk-of-poverty threshold, which is set at 60% of the national median equivalized disposable income (after social transfers).

Factor 14 by Eurostat definition: The indicator measures the share of severely materially deprived persons who have living conditions severely constrained by a lack of resources. They experience at least 4 out of 9 following deprivations items: cannot afford (i) to pay rent or utility bills; (ii) keep home adequately warm; (iii) face unexpected expenses; (iv) eat meat, fish or a protein equivalent every second day; (v) a week holiday away from home; (vi) a car; (vii) a washing machine; (viii) a color TV; or (ix) a telephone. The indicator is part of the multidimensional poverty index.

Factors from 15 to 24 by Eurostat definition: National accounts are a coherent and consistent set of macroeconomic indicators, which provide an overall picture of the economic situation and are widely used for economic analysis and forecasting, policy design, and policy making. These factors are from breakdowns of GDP aggregates and employment data by main industries and asset classes, employment by A*10 industry breakdowns Factors are segmented by an activity sector.

Factors 25 to 29 by Eurostat definition: The Human Resources in Science and Technology factors provides data on stocks and flows (where flows in turn are divided into job-to-job mobility and education inflows). The data on stocks and job-to-job mobility are obtained from the European Union Labour Force Survey (EU LFS). The National Statistical Institutes are responsible for conducting the surveys and forwarding the results to Eurostat.

Data on education inflows are obtained from Eurostat's Education database and in turn obtained via the UNESCO/OECD/Eurostat questionnaire on education.

Factors 3–5 are part of the indicator sets:

(a)　EU Sustainable Development Goals (SDG) indicator set where it is used to monitor progress towards SDG 9 on industry, innovation and infrastructure. SDG 9, among other things, recognizes the importance of technological progress and innovation for finding lasting solutions to social, economic and environmental challenges such as creating new jobs and promoting resource and energy efficiency.

(b)　EU 2020 strategy indicators where it is used to monitor progress towards the EU's target of 'improving the conditions for innovation, research and development', in particular with the aim of 'increasing combined public and private investment in R&D to 3% of GDP' by 2020.

Indicator can be considered as a global SDG indicator 9.5.1 "Research and development expenditure as a proportion of GDP". Furthermore, the indicator is part of the impact indicators for Strategic plan 2016–2020, referring to the 10 Commission priorities. Research and development (R&D) and innovation are key policy components of the Europe 2020 strategy. Innovative products and services not only contribute to the strategy's smart growth goal but also to its inclusiveness and sustainability objectives. Introducing new ideas to the market promotes industrial competitiveness, job creation, labor productivity and the efficient use of resources.

Factors 6 and 7 are part of the indicator sets:

(a)　EU Sustainable Development Goals (SDG) indicator set where it is used to monitor progress towards SDG 4 on ensuring inclusive and quality education for all and SDG 5 on gender equality. SDG 4 seeks to ensure people have access to equitable and quality education through all stages of life, from early childhood education and care, through primary and secondary schooling, to technical and vocational training, and tertiary education. SDG 5 aims at achieving gender equality by, among other things, ending all forms of discrimination, violence, and any harmful practices against women and girls in the public and private spheres.

(b)　EU 2020 strategy indicators is used to monitor progress towards the EU's target of 'increasing the share of the population aged 30 to 34 having completed tertiary or equivalent education to at least 40%' by 2020.

Furthermore, the indicator is part of the impact indicators for the strategic plan 2016–2020, referring to the 10 Commission priorities and included as a secondary indicator in the Social Scoreboard for the European Pillar of Social Rights.

Education and training lie at the heart of the Europe 2020 strategy and are seen as key drivers for growth and jobs. The EU has defined upper secondary education as the minimum desirable educational attainment level for EU citizens. People with a low level of education may not only face greater difficulties in the labor market but also have a higher risk of poverty and social exclusion. At the same time, education and training help boost productivity, innovation, and competitiveness.

### 2.3. Data Modification

Within the individual factors, ranking countries in individual years was determined. In the case of conformity, the weight of the country was used, and calculated according to size and population. Subsequently, individual factors and negative and positive factors were summarized for the period.

The group of EU countries was adjusted on the basis of available data and extended in the first phase to countries that may have an influence on the European countries in the significance of the selected factors, candidate countries, respectively other countries (customs union, monetary agreement, Central European Free Trade Area, etc.): Bosnia and Herzegovina, Northern Macedonia Serbia, Montenegro, Albania, and Turkey. Based on the systemic approach, two neighborhoods are defined, the first being made up of countries neighboring some of the selected countries (Russia, Ukraine, Belarus), and the second being made up of seas or oceans that may form an indirect border.

Based on the available data, countries where data recovery was less than 30% were finally excluded (Albania, Bosnia and Herzegovina, Iceland, Liechtenstein, Montenegro). For the remaining countries, in the case of incomplete data in the time series, the missing data in the relevant year were interpolated from the previous and the next known year. After data addition, 50% data recovery was tested, and Serbia and Turkey were excluded from data processing. Norway and North Macedonia (76%) and Switzerland (79%) also had a very small data base too. The missing series were supplemented by extremes (>max, <min) in these countries, according to the missing factor's positivity. Due to the comparative analysis, these states do not affect other values, they do not have real defined precedents, nor was the weight of the state was used according to its population and area. For Russia and Turkey, there was a reduction in significance according to the share of the European continent (decrease in value of area and population in proportion to the Asian and European parts).

Due to the generated infrastructure, a group of excluded countries has been analytically (not structurally) added to the neighborhood, the statistical data has been replaced by average incoming country data on the structure in order to ensure the passage of agents in these countries from lower factor to higher. The final infrastructure was generated as a combination of physical boundaries and further by geographic coordinates and identification of nearby states regardless of physical neighborhood.

In this case, the links were identified by specifying the number of the nearest states and by allowing or disabling the rewriting of the minimum found. In case of, that min (A → B) = min (B → A), where A and B are any pair of states minimum found. In the case of being necessary to ensure the coherence of infrastructure it is possible to add a pair of identified edges by triangulation.

Individual countries were scaled depending on the factor's negativity or positivity. The scaling was based on ranking individual countries according to the value of the relevant factor. It was based on ranking individual countries according to the value of the relevant factor. Basic scaling is for individual years, then countries were rated for a given factor and year by order of each year.

The evaluation for individual factors in the interim period is determined by the sum of partial years. There were recorded increases or decreases in the valuation of individual states, based on the established order. Differences in ranking between these states were identified by comparing available of states with existing bind (according to the generated infrastructure). Identification was carried out by passing autonomous agents over the infrastructure in random order of the state. Subsequently, precedence matrices were generated for each factor and year, recording the direction of the increase in the monitored values (ranking of states in the given year and factor).

At the present stage of the research, the first precedence was calculated. In the case that the precedent of the analyzed state is a nearby state or a sea precedent, this precedence is not counted in the case of the first precedence. For multiple precedencies, the precedence is included in the calculation because it can serve as part of a longer path. If the surrounding state or the sea is the initial node of a longer journey, the node is not taken into account.

## 3. Results

The analysis was performed on a virtual infrastructure. This infrastructure ensures an even distribution of factors while accepting the geographical context. Spatial links are modified in the model, the model adjusts the reality so that the model accepts geopolitical relations (state borders), the model also accepts the geographical distribution of countries (distances between countries, density of countries in the area).

Gradients of factor values and factor groups are monitored using this virtual infrastructure. Gradients identify local extremes for the respective combination of factors.

### 3.1. Virtual Infrastructure

Modeling infrastructure was generated in the first part of the analyzes. The default structure was created by a physical neighborhood (boundary). The structure was completed by generating a pair of minimum distances between states.

Eliminating pairs of identical identified edges between pairs of states was performed by generating one edge without transcription and adding triangulation (Figure 1 shows the generation of infrastructure). Blue is the initial fragment that has been generated by generating one edge with forbidden repetition. Green is represented by edges created by adding two minima with the possibility of rewriting. The physical border infrastructure (left side), including links to neighborhood 1 and neighborhood 2 (sea and neighboring states), is marked in red.

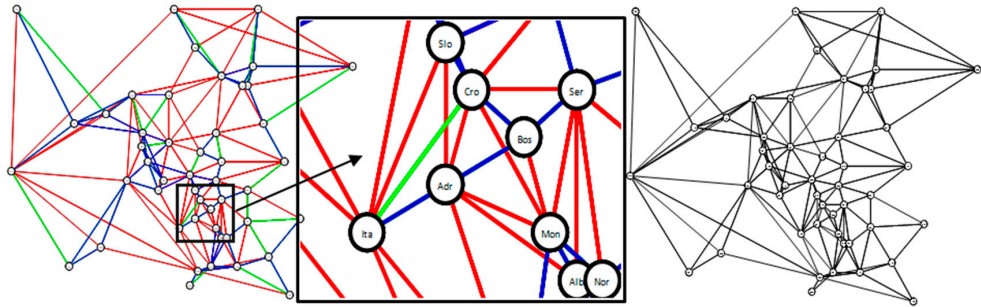

**Figure 1.** Infrastructure used for the analysis.

The middle part of the picture shows a detailed view, where the connection between Italy and Bosnia through the Adriatic Sea (Slo—Slovenia, Cro—Croatia, Ser—Serbia, Bos—Bosnia and Herzegovina, Ita—Italy, Alb—Albania, Nor—North Macedonia, Mon—Montenegro, / neighborhood 2 /, Adr—Adriatic Sea / neighborhood 1 /).

The right part shows generated infrastructure for autonomous systems (black). This infrastructure is used for other analysis. In the next research phase the values of individual factors were compared and weights were assigned. The analysis was based on factors, groups and effects (see Table 2). The effect is positive or negative, depending on the effects of Industry 4.0 (for example, a higher share of the IT industry is a positive factor and a higher share of the service industry is a negative factor).

Weights have been assigned so that a higher score means more benefit. Consequently, the order is reversed, for negative factors. Therefore, more points mean less risk (threat).

The graph in Figure 2 shows that several groups of states can be identified for the selected factors. Greece, Portugal, Norway, Italy, Hungary, Switzerland, Latvia, Estonia, Spain, Cyprus, and Lithuania. For all these countries, the total score for negative factors is higher than positive factors.

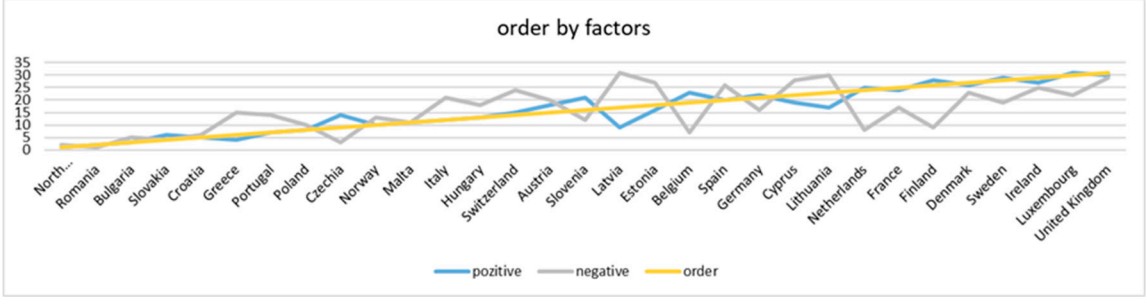

**Figure 2.** Factors—rank.

The group with fewer negative points includes Slovakia, Czechia, Slovenia, Belgium, Germany, the Netherlands, France, Finland, Denmark, Sweden, Ireland, Luxembourg, and the UK. The groups of states with the lowest threat (most points) are more influenced by positive factors. (Netherlands, France, Finland, Denmark, Sweden, Ireland, Luxembourg, UK).

The graph also shows that the countries with the highest difference between negative and positive factors and a higher proportion of negative factors are Greece, Latvia, Estonia, and Lithuania, which are countries with generally worse economic potential. Switzerland has a similar ratio.

### 3.2. Analysis Using Precedence

The next phase of the analysis was to calculate the precedents for individual countries for individual factors. Almost 20,000 first precedents and 355 long precedents for individual states and factors in individual years were identified in the period under review. Table 4 shows the numbers of precedencies by factor for all countries under review. The smallest number of short precedents was in the 'employment' factor, where it was at a minimum (see Table 4). The exceptions were 2010 and 2011 when the number of precedents was slightly higher (2010–74, 2011–73). The 'agriculture' factor (values 12–73) and the 'material deprivation factor (values 72–74) were also below the average. Above average factors are 'real estate activities', 'professional, scientific and technical activities; administrative and support service activities', 'public administration, defence, education, human health and social work activities', 'Arts, entertainment and recreation; other service activities', 'HRST', 'HRSTE', 'HRSTO', 'HRSTC', and 'SE' (approx. 76-77 precedencies).

**Table 4.** First and long precedencies by factors.

| First | | | | Long | | | |
|---|---|---|---|---|---|---|---|
| **Min.** | **Max.** | **Avg.** | **Med.** | **Min.** | **Max.** | **Avg.** | **Med.** |
| 70 | 81 | 76 | 76 | 6 | 19 | 13 | 13 |

The 'employment—women' factor showed the most short precedencies, where the number of precedents ranged from 80 to 81. For long precedents, minimum values were recorded in 2013 for the 'real estate activities' factor (6). Maximum precedence rates were identified in 2014 for the 'material deprivation' factor (19). Interestingly, the values distribution is the same as the median for both long and first precedencies.

Short precedencies in the European context must show the same numbers because for each pair of states with detention it has always one higher value of the given factor. The number of precedents then shows the ratio to the surroundings. Therefore, it is evident that employment in the area of the monitored countries shows a higher share in relation to the surrounding area than in neighboring countries. It can also be seen that the 'material deprivation' factor is lower in the countries under review. The increase in long precedence values and their decline in 2013 for 'material deprivation' shows the depletion of funds after the crisis. A considerable number of long precedents indicate stabilization in individual countries, with neighboring countries showing little differentiation. On the contrary, the minimum of long precedents in 'real estate activities' indicates an increased number of local extremes, which is caused by an increased differentiation of states in this factor. Table 5 shows an overview of the maximum and average values and the median precedence values by country.

**Table 5.** First and long precedencies by states.

| First | | | | Long | | | |
|---|---|---|---|---|---|---|---|
| **Min.** | **Max.** | **Avg.** | **Med.** | **Min.** | **Max.** | **Avg.** | **Med.** |
| 0 | 1193 | 5,088,125 | 493 | 0 | 71 | 1,109,375 | 5.5 |

Malta has the first lowest precedents, with Spain (220), Portugal (198) Norway (193), and Cyprus (168) showing the lowest values. The graph in Figure 2 shows that these are mainly countries where the importance of negative factors prevails. Austria (1193), France (1056), and Luxembourg (1015) have the maximum number of first precedents.

These countries have the best results, taking into account the geographical aspect, and comparison of the factor values to the immediate surroundings. For long precedencies, Malta, Latvia, and Portugal show zero precedents for all factors over the entire reporting period. Luxembourg (71), Switzerland (39), and Ireland (37) have the longest precedents. The chart on the right in Figure 3 shows that these countries make up almost half of the precedents (along with North Macedonia).

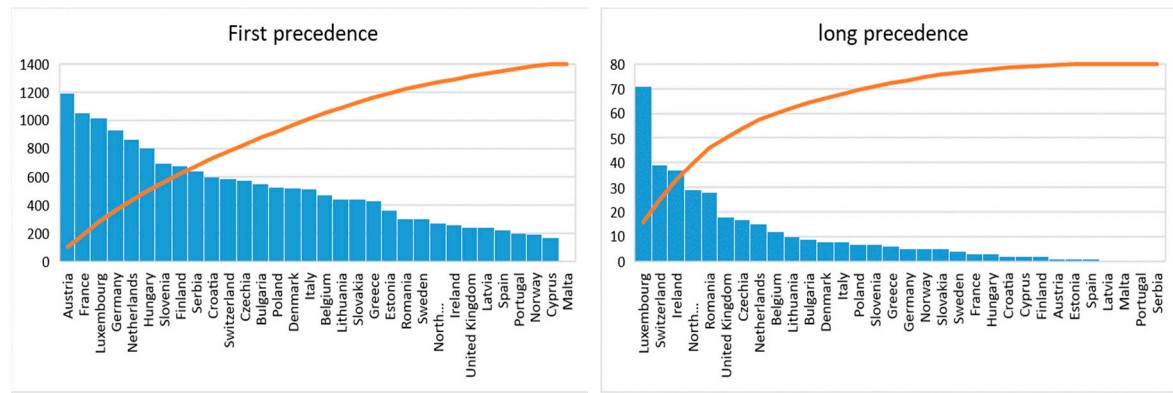

**Figure 3.** Total number of precedencies.

For Luxembourg and Ireland, the number of long precedents corresponds to the values according to the factors, for Switzerland the number is influenced by geographical location.

Figure 4 shows the geospatial distribution of the first (left) and long (right) parts. Darker color means higher values.

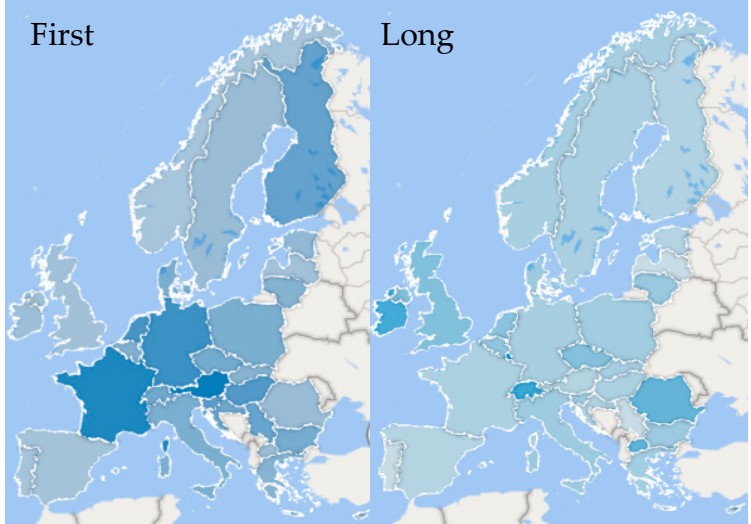

**Figure 4.** Number of precedencies by country—geospatial distribution.

Long precedents indicate that there are sequences of factor values around the state with a slightly decreasing value, but with a greater geographical impact. Short precedencies indicate a large number of neighboring states in the immediate vicinity with lower factor values. In other words, a long precedence indicates that it is not an isolated (to a certain distance that is less than the length of the long precedence) maximum.

If we compare the development according to the changes in the given time interval, in 2010 and 2018, then we can see that the situation is relatively stable. Table 6 shows the basic statistical values. For the first precedents, Luxembourg (122) has the maximum precedence in 2010, followed by France (119) and Austria (118).

**Table 6.** First and long precedencies by years.

| | 2010 | | | | | | | | 2018 | | | | | | |
|---|---|---|---|---|---|---|---|---|---|---|---|---|---|---|---|
| | **First** | | | **Long** | | | | **First** | | | | **Long** | | | |
| **Min.** | **Max.** | **Avg.** | **Med.** | **Min.** | **Max.** | **Avg.** | **Med.** | **Min.** | **Max.** | **Avg.** | **Med.** | **Min.** | **Max.** | **Avg.** | **Med.** |
| 0 | 122 | 56.5 | 52.5 | 0 | 8 | 1.25 | 1 | 0 | 143 | 56.5 | 53 | 0 | 8 | 1.06 | 0 |

In 2018 there was a significant increase in Austria (143, max), with high values in France (118), Luxembourg (105), Germany (103), and the Netherlands (100). In both cases, the peak values are above twice the mean, the median in both years being smaller than the mean.

A comparison of the first precedents of all countries is shown in Figure 5. The relative stability in the order of countries with a large number of first precedents is apparent.

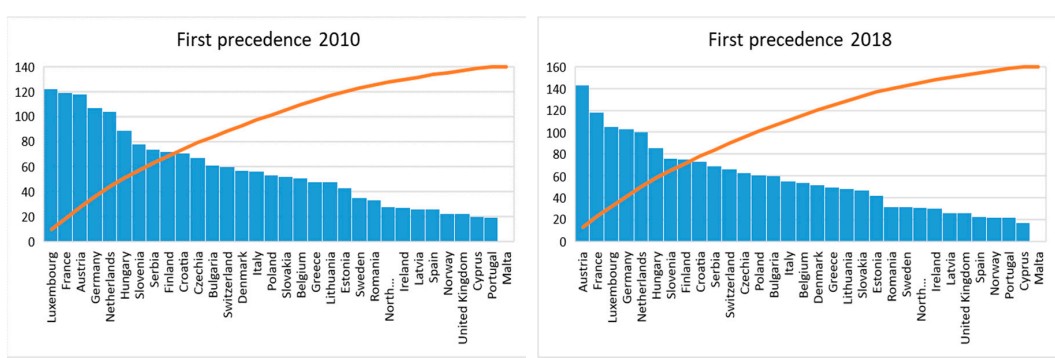

**Figure 5.** Number of the first precedencies by country in 2010 and 2018—geospatial distribution.

There is a significant increase in the number of first precedents in Austria (118 → 143), indicating an increase in values in the sum of all factors and a marked increase in the country's dominance in the region.

This is surprising, especially when compared to standard dominant countries such as France and Germany.

Development in the number of long precedencies is shown in Figure 6. In both years monitored, the value of Luxembourg's maximum is evident (eight precedents). However, in 2018 this dominance is no longer so pronounced.

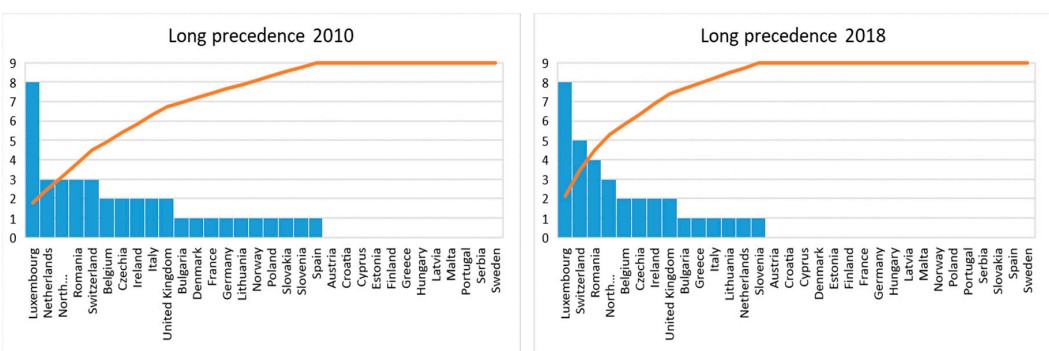

**Figure 6.** Number of the long precedencies by country in 2010 and 2018—geospatial distribution.

It is also evident that the number of states with the longest precedents decreased in 2018 (from 20 to 14). Luxembourg, Romania and Switzerland form a significant number of long precedents in both years. As can be seen from Figure 6, this share in 2018 is higher than 50 percent of the total number of long precedents.

This means that in 2018 local extremes increased in the vicinity of most countries, which do not allow the creation of long precedents. Thus, the uniform distribution of factor values in space has been reduced and disproportion between countries in a geographically close neighborhood has increased.

The graph in Figure 7 shows that Austria's first precedence increases equally and gradually (except for 2017), while Luxembourg also declines gradually and evenly. For other countries a steady state is apparent. The Netherlands returned to their original values after a slight decline, Estonia and Greece have similar developments. A slight increase is shown in Belgium (2010–2017) and Poland (growth in 2011–2013, then stagnation).

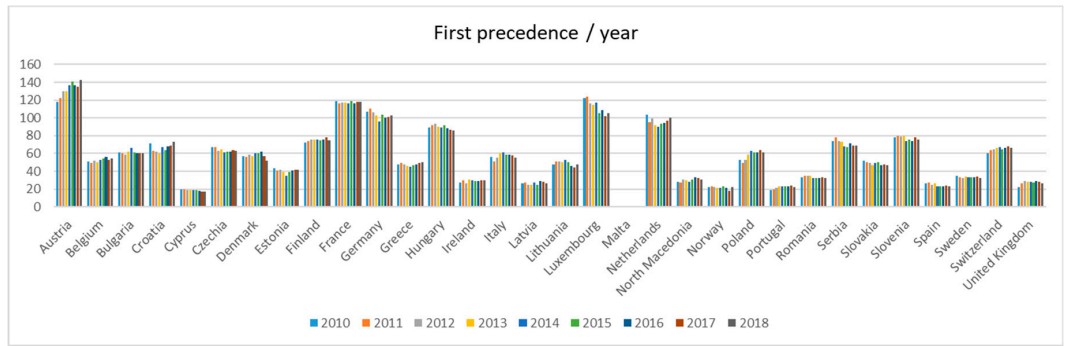

**Figure 7.** Number of the long precedencies by country, 2010–2018—geospatial distribution.

The long precedence (Figure 8) shows a large number in Ireland, where the high precedence rates are in 2011–2017, the boundary years of the interval are lower, so they have not been shown in the Figure 6. North Macedonia also has a high precedence.

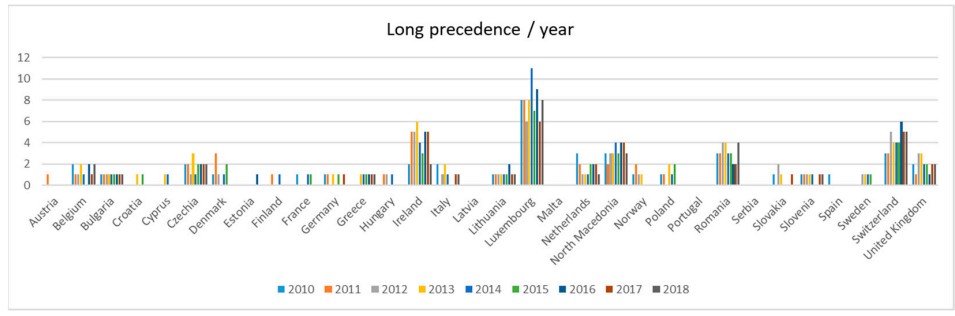

**Figure 8.** Number of long precedence by country, 2010–2018—geospatial distribution.

For long precedencies, the volatility of the values is evident, indicating that even if there is stability in the first precedents, states with high numbers of first precedencies are not always local maxima.

The spatial layout is apparent from Figure 9. They are depicted progressively from left to right: first precedencies 2010, first precedencies 2018, and Figure 10, long precedencies 2010, long precedencies 2018. The darker color shows higher numbers of precedence.

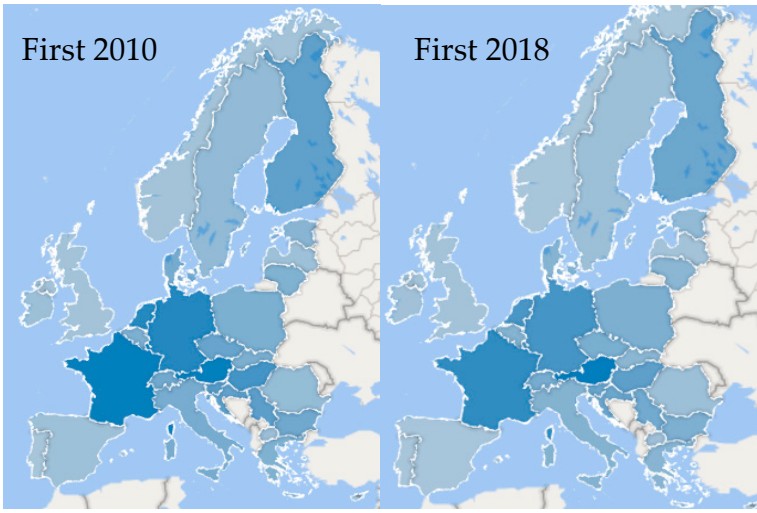

**Figure 9.** Number of first precedencies by country—geospatial distribution in 2010 and 2018.

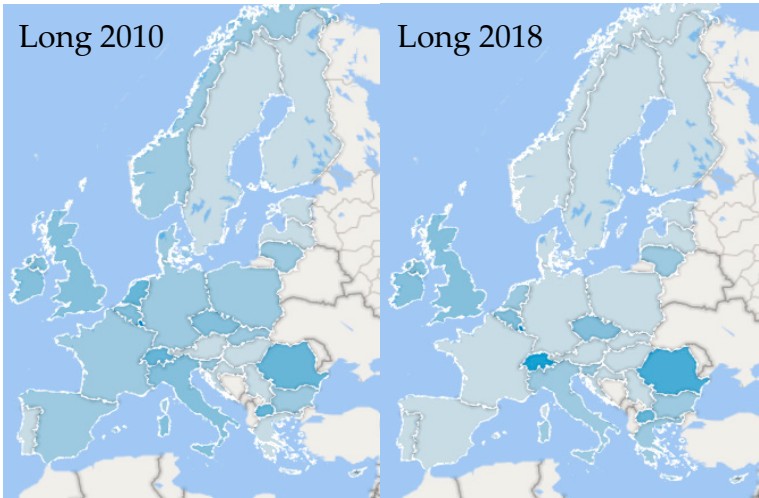

**Figure 10.** Number of long precedencies by country—geospatial distribution in 2010 and 2018.

### 3.3. Summary by Group

The next stage was to compare precedencies by individual factors. The sub-factors were further grouped by priority area of influence into RD, Em, Soc, Edu, and Eco (see Dates and Methods) and by action positivity and negativity. Table 7 shows basic statistical data by group, further broken down by positive and negative factors.

**Table 7.** First and long precedencies by years.

| Effect | Precedence Group | Long Eco | First Eco | Long Edu | First Edu | Long Soc | First Soc | Long RD | First RD | Long Em | First Em |
|---|---|---|---|---|---|---|---|---|---|---|---|
| all | min | 0 | 0 | 0 | 0 | | | | | 0 | 0 |
| all | max | 27 | 478 | 11 | 232 | | | | | 9 | 79 |
| all | pr | 5.03 | 228 | 2 | 87.6 | | | | | 0.6 | 35 |
| all | med | 1.5 | 234 | 0.5 | 83 | | | | | 0 | 33.5 |
| positive | min | 0 | 0 | 0 | 0 | | | 0 | 0 | 0 | 0 |
| positive | max | 9 | 130 | 9 | 114 | | | 44 | 318 | 4 | 61 |
| positive | pr | 1.25 | 52.9 | 0.7 | 35 | | | 2.4 | 105 | 0.3 | 17.3 |
| positive | med | 0 | 46 | 0 | 33.5 | | | 0 | 91.5 | 0 | 13 |
| negative | min | 0 | 0 | 0 | 0 | 0 | 0 | | | 0 | 0 |
| negative | max | 27 | 382 | 6 | 118 | 164 | 13 | | | 9 | 45 |
| negative | pr | 3.8 | 175.4 | 1.3 | 52.7 | 52.8 | 1 | | | 0.3 | 17.7 |
| negative | med | 1 | 176.5 | 0 | 44 | 42.5 | 0 | | | 0 | 18 |

3.3.1. Group Eco

In the Eco Group, the most precedents are in Germany (478), France (423), and Luxembourg (420), with an above average median. In summary, figures for the first precedents are shown in Figure 11. Especially for Luxembourg (130, max) and Hungary (109), it is clear that it has greater potential in the positive factors segment, while Austria has a higher share of negative factors (321). France has a large share in both the positive and negative (123) segment. Germany (382) has the maximum number of negative factors and Luxembourg (130) has positive precedence. Norway, Portugal, Spain, and Cyprus have the lowest number of cases. Malta has no first or long precedencies.

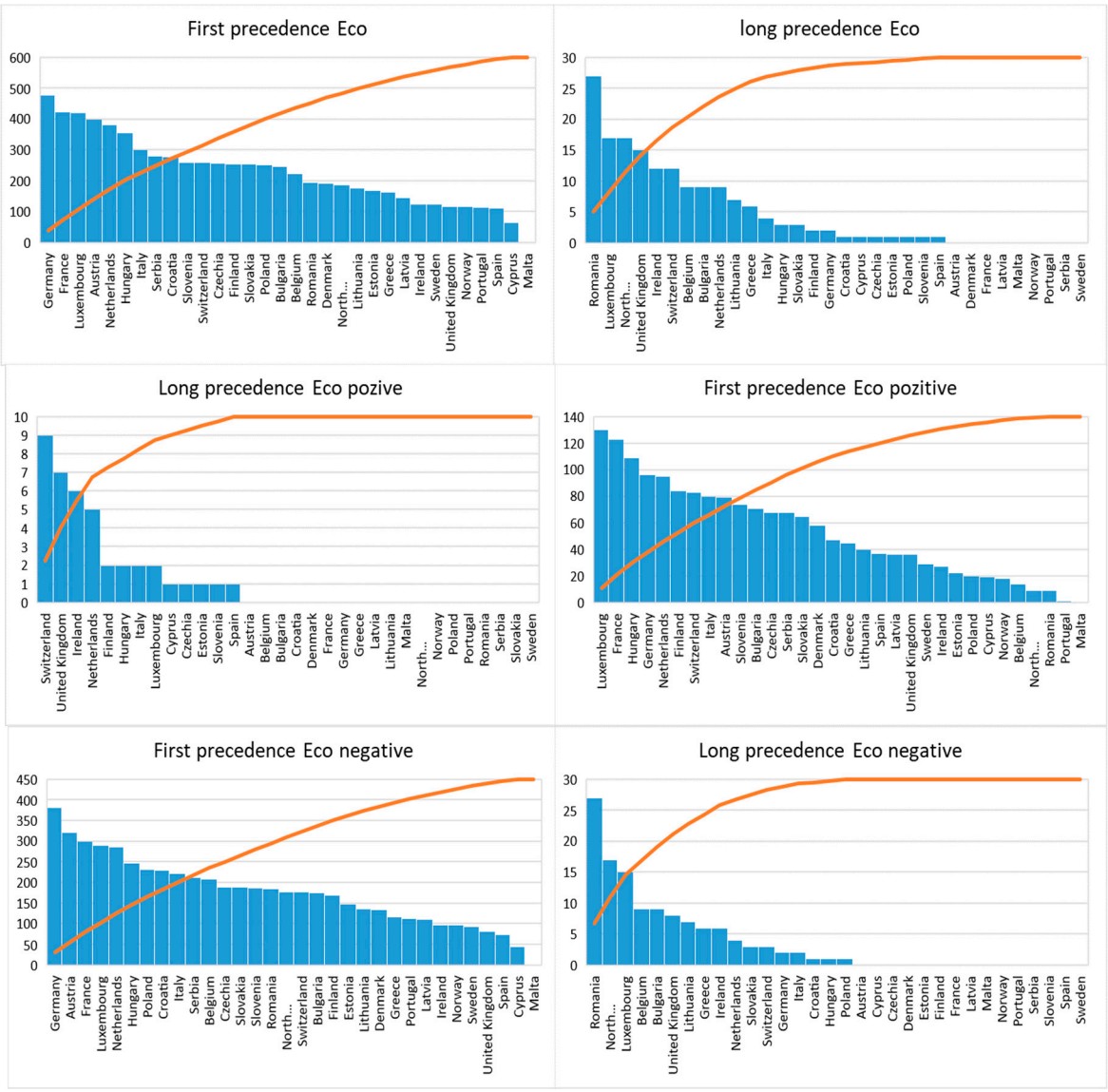

**Figure 11.** Number of precedence by country, first and long precedency groups Eco, by effect.

Figure 11 shows that Romania (27) shows the longest precedencies, Switzerland (9), UK (7), and Ireland (6) in the positive factors segment. Romania also has the highest number of long precedents for negative factors (27), and North Macedonia also has a high number (17). This distribution is due to the neighborhood, which circumvents local extremes, especially in the center of Europe. In all cases, the mean significantly exceeds the median. The geospatial distribution is on Figures 12 and 13,

first figure shows the first precedence in the order from left to right: all, positive, negative, second figure shows the long precedence in the same order.

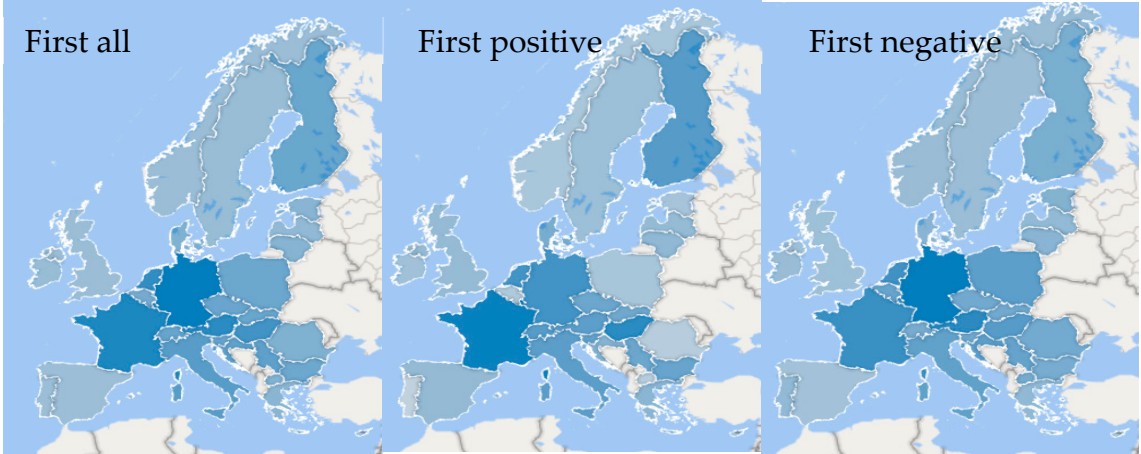

**Figure 12.** Number of precedence by country, first precedencies group Eco, by effect. Geospatial distribution.

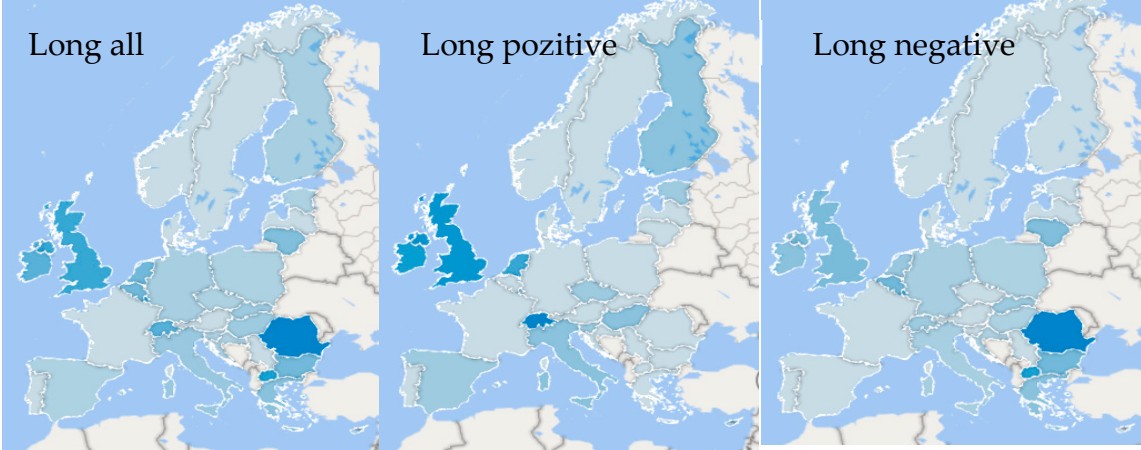

**Figure 13.** Number of precedence by country, long precedencies, group Eco by effect. Geospatial distribution.

3.3.2. Group Edu

This group of attributes reaches the maximum number of Austria (232), and a high numbers are identified in France (181) and Switzerland (177). Austria also shows high numbers of first precedents for positive Edu factors (max, 114) and France (80). Hungary also has a high number (73). Figure 14 shows the distribution of Edu Factor Group values by country, as in Figure 11. Similarly, Figures 15 and 16 has the distribution as Figures 12 and 13.

Austria has the highest number of first precedents also in the group of negative Edu factors (118, max); furthermore, it shows high values of Netherlands (116), Croatia (106), and France (101).

For long precedents, there is a high number of precedents in Switzerland (max 11), followed by Luxembourg (9) and Ireland (8). In the positive group of Edu factors, the maximum number is in Switzerland (9), in the negative group of Edu factors the maximum number is Poland and Luxembourg (6), high values of Edu negative precedence are also in Italy (4).

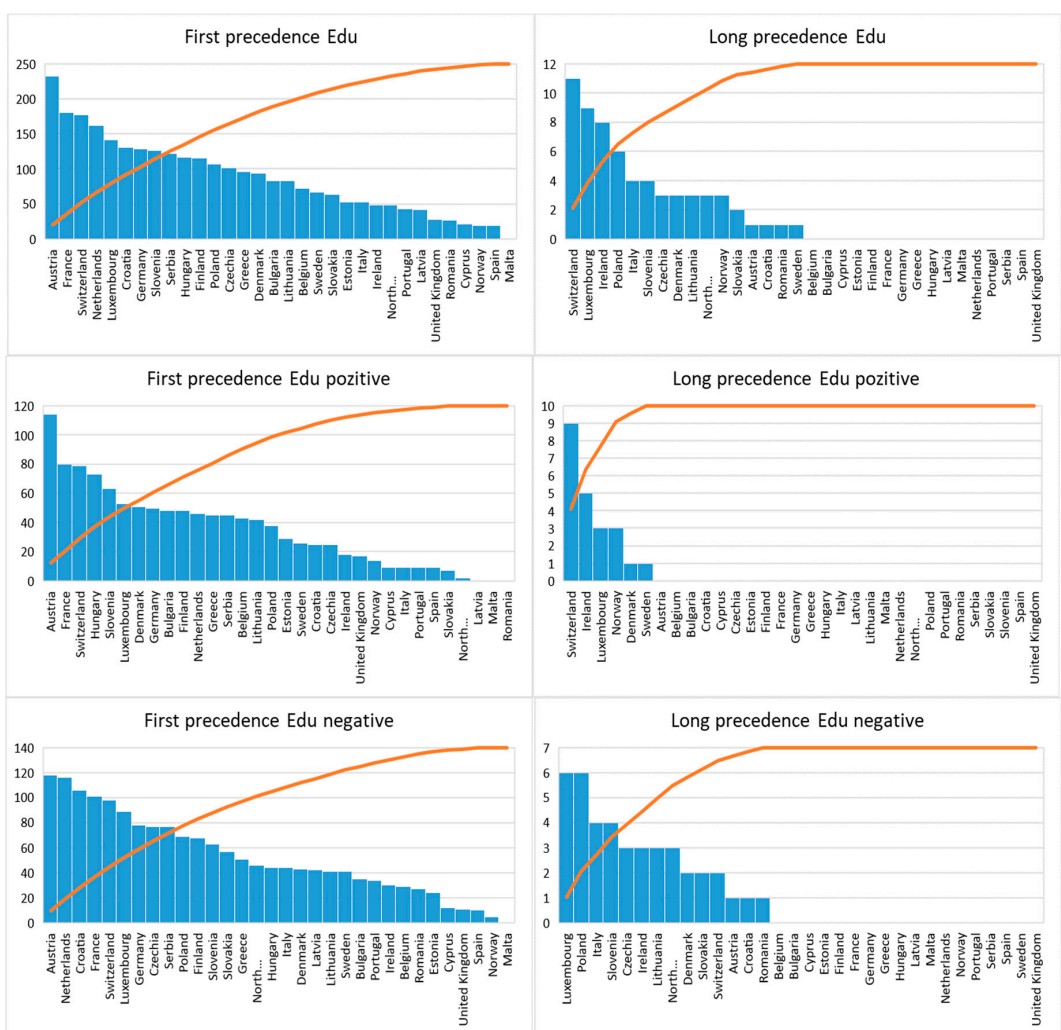

**Figure 14.** Number of precedence by country, first and long precedencies, grouped Edu by effect.

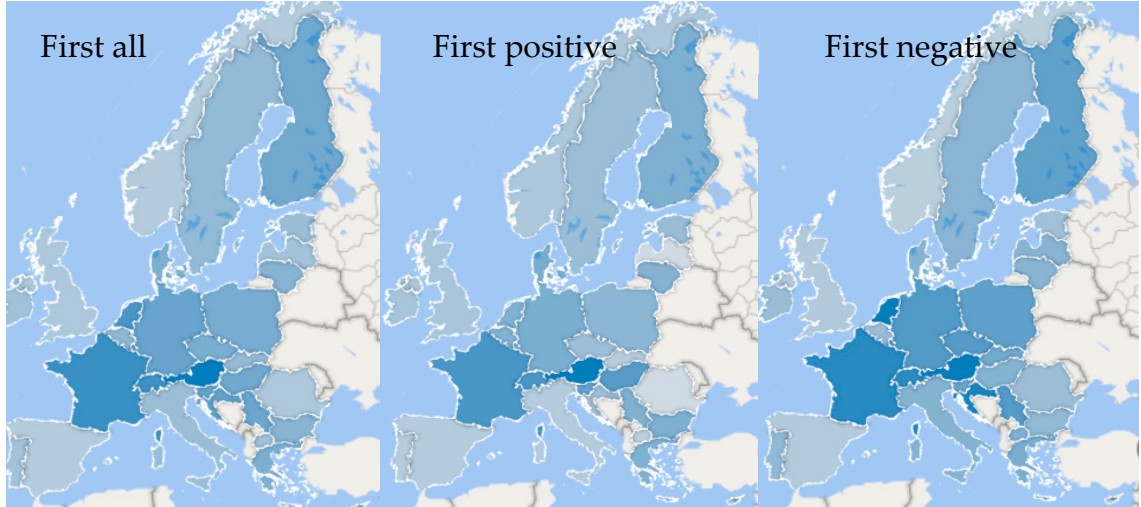

**Figure 15.** Number of precedence by country, first precedencies, group Edu by effect. Geospatial distribution.

A relatively surprising finding is the existence of only six long precedents for positive factors. This indicates a high number of local maxima, but in the case of Switzerland it is not a long precedence

due to the geographical neighborhood with the surroundings. In all groups the mean is higher than the median, there is a relatively large difference in the first precedents of negative factors (52.7 → 44). This ratio is given by a relatively large group of countries with a high number of first precedents. Austria, Netherlands, Croatia, France, Switzerland, Luxembourg, Germany, and Czechia are above average. An interesting finding is the position of Croatia, because its economic character does not fit among the other identified countries.

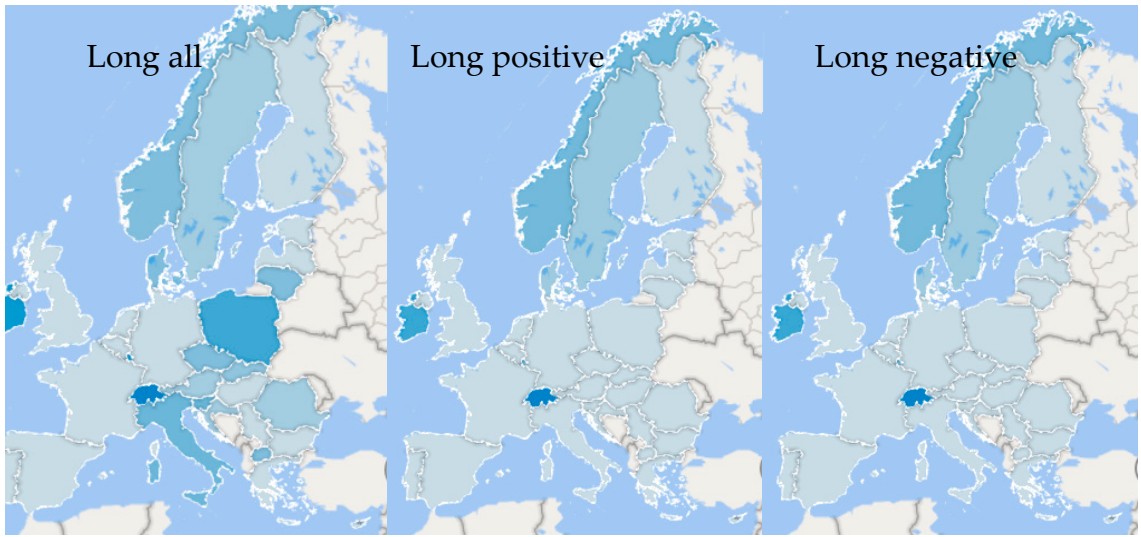

**Figure 16.** Number of precedence by country, long precedencies group Edu by effect. Geospatial distribution.

### 3.3.3. Group EM

When examining the precedence of the employment factors group, it was identified that the maximum of short precedents for that group was reached by Austria (79), with a relatively large number from Germany (74), Luxembourg (64) and the Netherlands (60). In the central part of Europe, which may result in the formation of isolated local maxima. Due to the fact that the mean is slightly above the median (35 → 33.5, see Table 4), it is clear that the number of longer precedents is slightly higher in Austria and Germany, in other countries it has a decreasing, relatively uniform character (Figure 17).

In the segment of positive EM factors, Germany has the largest share (61), Austria is in second place with (59). For this group, the average is much higher than the median (17.4 → 13), Figure 17 shows that almost 50% of the first precedents are Germany, Austria, Switzerland, Netherlands, and France, the following location of the Bulgaria, Hungary, Slovenia, and Estonia. However, Bulgaria is very poorly ranked in the group of negative EM factors, where together with Cyprus, Malta, and Switzerland they have no precedence, which means that they have lower values in this group of factors than neighboring states.

Italy has the highest precedence in the group of negative EM factors (45), while Luxembourg has a slightly smaller number of precedencies (43). These countries follow quite a long distance between Belgium (36), Poland (33), and Croatia (33), see Figure 17. The mean and median differ only slightly (17.7 → 18)

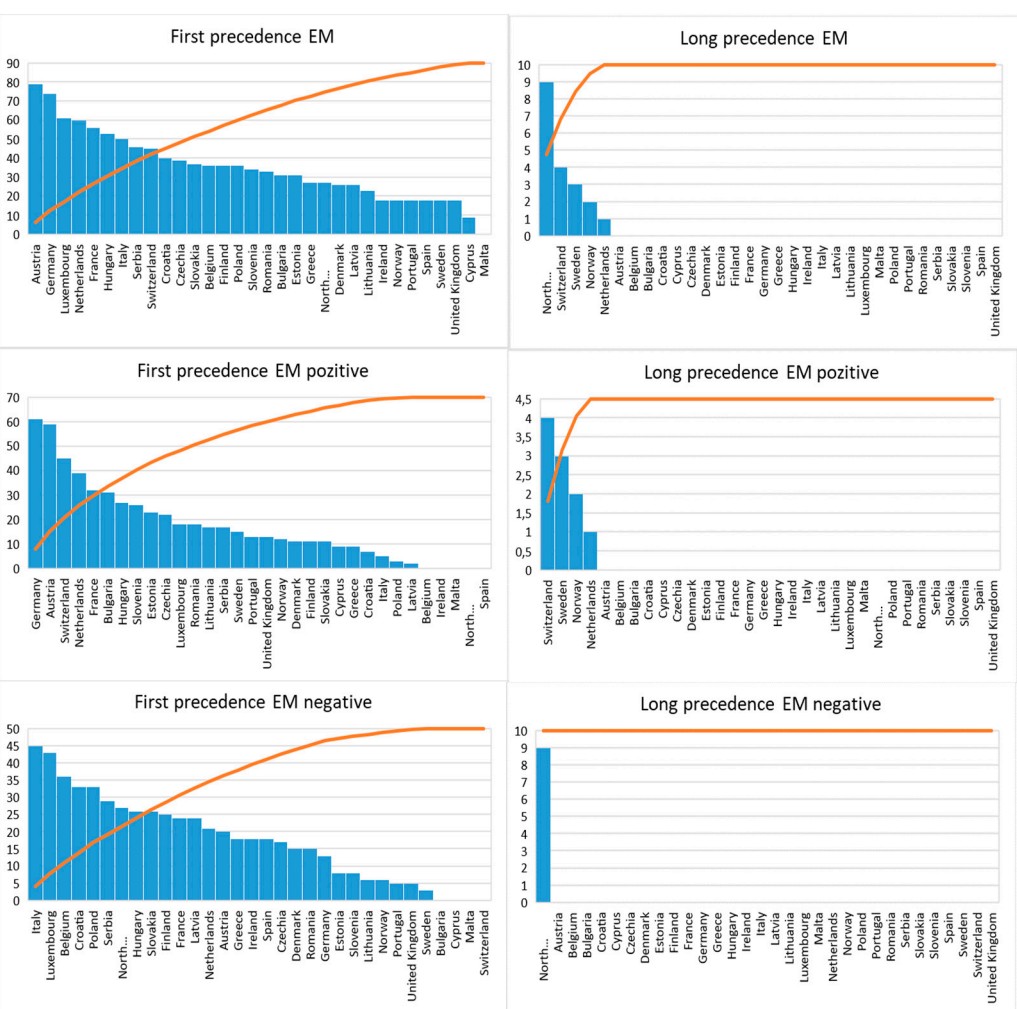

**Figure 17.** Number of precedence by country, first and long precedence group EM by effect.

For long precedents, a relatively small number of states are identified in this group. There are five states for the EM group, four states for the positive EM factors, and three states for the negative EM factors, as evidenced by the zero median value in these groups (Table 4). For the EM group, the maximum number of precedents is North Macedonia (9), the same country has a maximum for the negative EM factor group (9), the only country with long precedents in this group. This indicates that there are more isolated local maxima for negative EM factors.

A small number of long precedencies does not always mean isolated local maxima (this is true for all factors and analyses), a low number of these precedencies may also be given by looking for the longest precedencies separately for each factor and group of factors. Long precedence for one factor is shorter than long precedence another factor. Switzerland (4, max), Sweden (3), Norway (2), and Netherlands (1) have long precedents for positive EM factors. Localization to the northern part of Europe is important. The visualization in the form of maps is in Figures 18 and 19, the maps are again in the same order as in the previous cases.

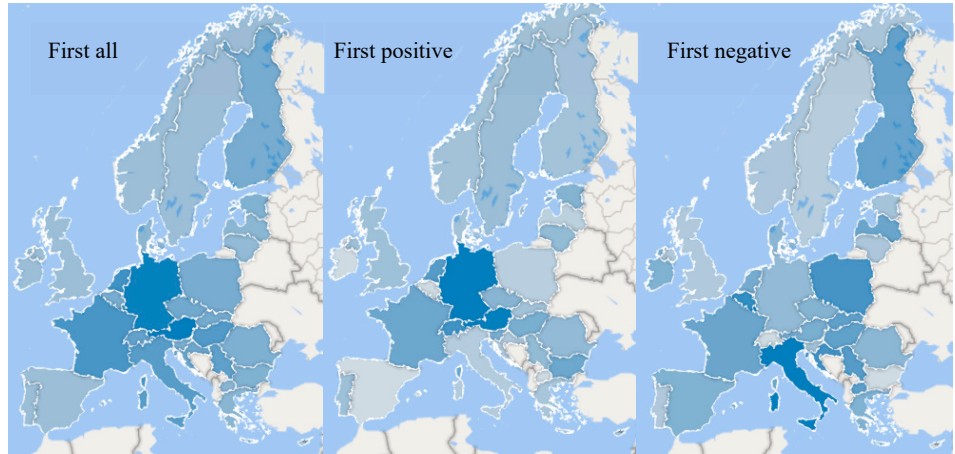

**Figure 18.** Number of precedence by country, first precedencies, group EM by effect. Geospatial distribution.

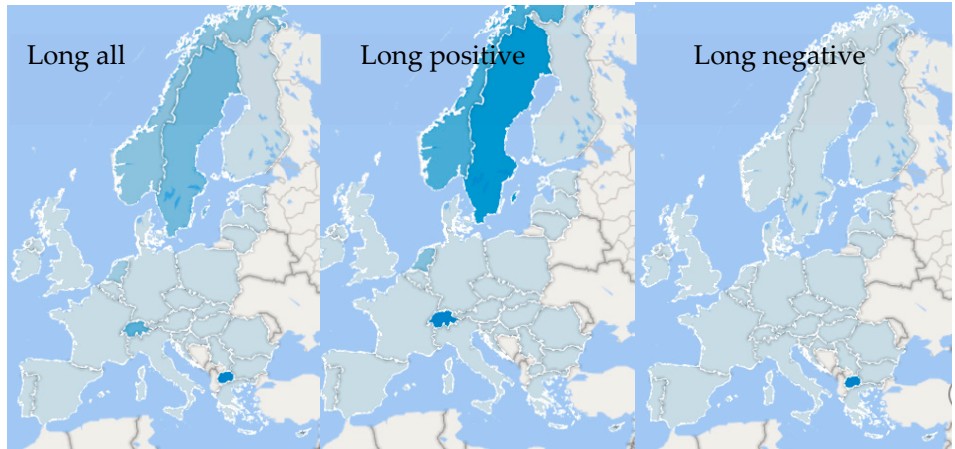

**Figure 19.** Number of precedence by country, long precedencies, group EM by effect. Geospatial distribution.

### 3.3.4. Group RD

There is no negative factor in this group, given the availability of statistical data and the fact that in most cases science and research will have a positive impact on Industry 4.0. The first precedents show a relatively uniform development (decline), above which Austria (318, max) and Luxembourg (308) are, with a slight distance are France, Hungary, Finland, Germany, and Slovenia. The presence of Slovenia in this group is surprising and shows the country's successful development in this area. Similarly, the same can be said for Bulgaria (Figure 20).

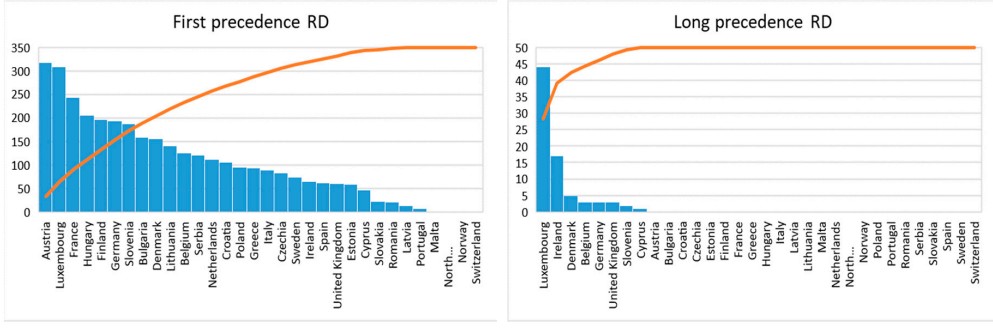

**Figure 20.** Number of precedence by country, first and long precedencies group RD.

Long precedents have been identified in eight countries, with more than half in Luxembourg (44, max), the second Ireland having 117 precedents. Denmark, Belgium, German, UK, Slovenia, and Cyprus also have long precedencies, these countries have five or less precedents. The precedencies on the map background are shown in Figure 21.

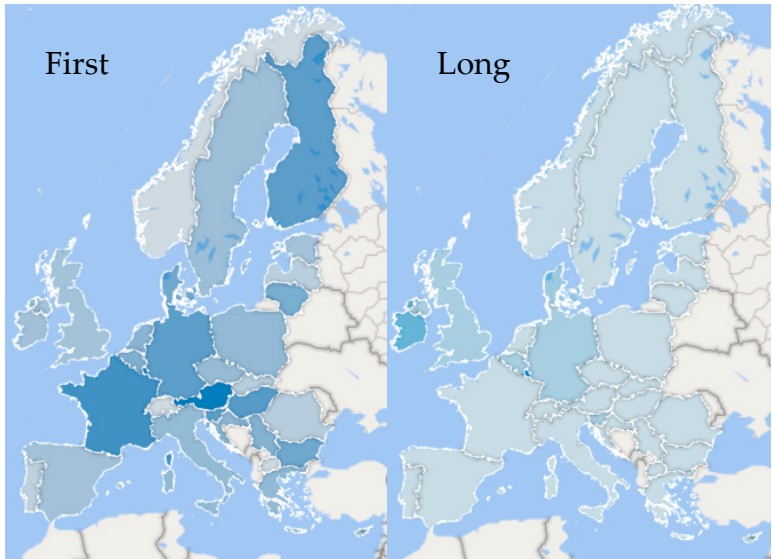

**Figure 21.** Number of precedence by country, first and long precedence group RD by effect. Geospatial distribution.

### 3.3.5. Group Soc

The group of Soc factors, unlike RD, has only negative factors, due to the fact that there are factors related to material deprivation and the risk of poverty in the group. The figures on Figure 22 show that Austria (164, max) has the highest first precedence, France (153) and the Netherlands also have a high number. High values in these countries cause the average to be higher than the median (52.8 → 42.5). Above-average values are also reported by Czechia and Slovakia, Czechia also has the longest precedents (13, max). Long precedents are found in only five countries, Czechia, Switzerland (12), Netherlands (5), France (3), and Luxembourg (1). Show by map is in the Figure 23.

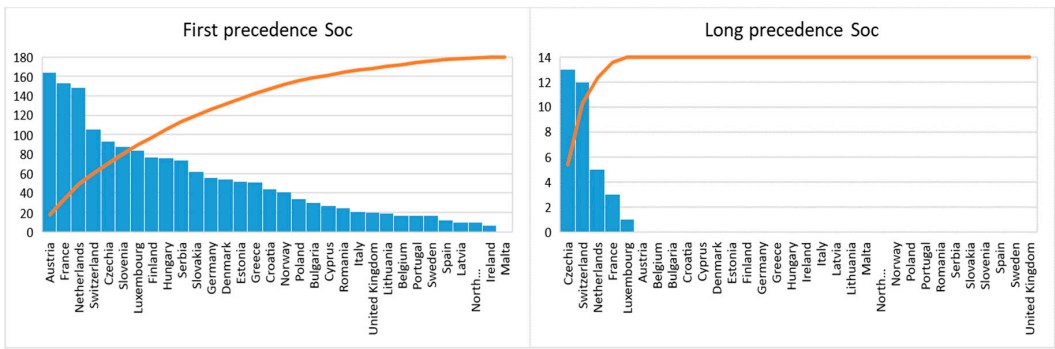

**Figure 22.** Number of precedence by country, first and long precedence group Soc.

Figure 24 summarizes the numbers of the first precedencies for each group of factors. The intensity of the red fill indicates the intensity of the country's dominance in a given set of factors.

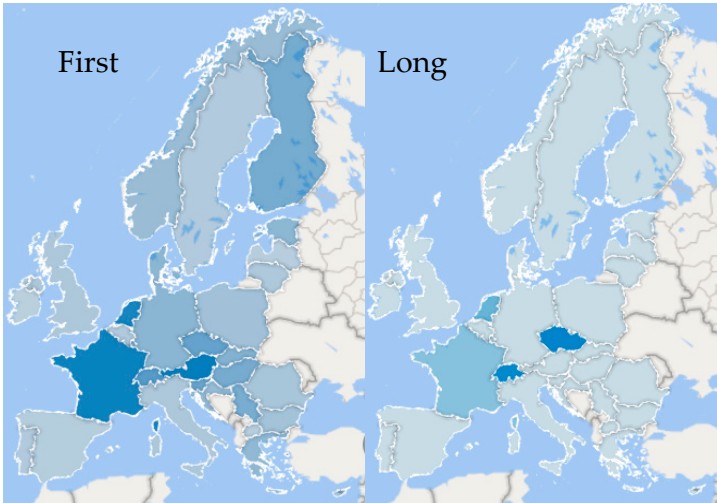

**Figure 23.** Number of precedence by country, first and long precedence group Soc by effect. Geospatial distribution.

| | | Austria | Belgium | Bulgaria | Croatia | Cyprus | Czechia | Denmark | Estonia | Finland | France | Germany | Greece | Hungary | Ireland | Italy | Latvia |
|---|---|---|---|---|---|---|---|---|---|---|---|---|---|---|---|---|---|
| ECO | all | 400 | 222 | 246 | 276 | 64 | 257 | 191 | 169 | 253 | 423 | 478 | 162 | 355 | 124 | 301 | 146 |
| ECO | positive | 79 | 14 | 71 | 47 | 19 | 68 | 58 | 22 | 84 | 123 | 96 | 45 | 109 | 27 | 80 | 36 |
| ECO | negative | 321 | 208 | 175 | 229 | 45 | 189 | 133 | 147 | 169 | 300 | 382 | 117 | 246 | 97 | 221 | 110 |
| EDU | all | 232 | 72 | 83 | 131 | 21 | 102 | 94 | 53 | 116 | 181 | 128 | 96 | 117 | 48 | 53 | 42 |
| EDU | positive | 114 | 43 | 48 | 25 | 9 | 25 | 51 | 29 | 48 | 80 | 50 | 45 | 73 | 18 | 9 | 0 |
| EDU | negative | 118 | 29 | 35 | 106 | 12 | 77 | 43 | 24 | 68 | 101 | 78 | 51 | 44 | 30 | 44 | 42 |
| EM | all | 79 | 36 | 31 | 40 | 9 | 39 | 26 | 31 | 36 | 56 | 74 | 27 | 53 | 18 | 50 | 26 |
| EM | positive | 59 | 0 | 31 | 7 | 9 | 22 | 11 | 23 | 11 | 32 | 61 | 9 | 27 | 0 | 5 | 2 |
| EM | negative | 20 | 36 | 0 | 33 | 0 | 17 | 15 | 8 | 25 | 24 | 13 | 18 | 26 | 18 | 45 | 24 |
| RD | positive | 318 | 125 | 159 | 106 | 47 | 83 | 155 | 59 | 196 | 243 | 194 | 94 | 205 | 65 | 89 | 14 |
| SOC | negative | 164 | 17 | 30 | 44 | 27 | 93 | 54 | 52 | 77 | 153 | 56 | 51 | 76 | 7 | 21 | 10 |

| | | Lithuania | Luxembourg | Malta | Netherlands | North Macedonia | Norway | Poland | Portugal | Romania | Serbia | Slovakia | Slovenia | Spain | Sweden | Switzerland | United Kingdom |
|---|---|---|---|---|---|---|---|---|---|---|---|---|---|---|---|---|---|
| ECO | all | 175 | 420 | 0 | 381 | 186 | 115 | 252 | 113 | 193 | 280 | 253 | 260 | 110 | 123 | 259 | 117 |
| ECO | positive | 40 | 130 | 0 | 95 | 9 | 18 | 20 | 1 | 9 | 68 | 65 | 74 | 37 | 29 | 83 | 36 |
| ECO | negative | 135 | 290 | 0 | 286 | 177 | 97 | 232 | 112 | 184 | 212 | 188 | 186 | 73 | 94 | 176 | 81 |
| EDU | all | 83 | 142 | 0 | 162 | 48 | 19 | 107 | 43 | 27 | 122 | 64 | 126 | 19 | 67 | 177 | 28 |
| EDU | positive | 42 | 53 | 0 | 46 | 2 | 14 | 38 | 9 | 0 | 45 | 7 | 63 | 9 | 26 | 79 | 17 |
| EDU | negative | 41 | 89 | 0 | 116 | 46 | 5 | 69 | 34 | 27 | 77 | 57 | 63 | 10 | 41 | 98 | 11 |
| EM | all | 23 | 61 | 0 | 60 | 27 | 18 | 36 | 18 | 33 | 46 | 37 | 34 | 18 | 18 | 45 | 18 |
| EM | positive | 17 | 18 | 0 | 39 | 0 | 12 | 3 | 13 | 18 | 17 | 11 | 26 | 0 | 15 | 45 | 13 |
| EM | negative | 6 | 43 | 0 | 21 | 27 | 6 | 33 | 5 | 15 | 29 | 26 | 8 | 18 | 3 | 0 | 5 |
| RD | positive | 141 | 308 | 0 | 112 | 0 | 0 | 95 | 7 | 21 | 121 | 23 | 187 | 61 | 74 | 0 | 60 |
| SOC | negative | 19 | 84 | 0 | 149 | 10 | 41 | 34 | 17 | 25 | 74 | 62 | 88 | 12 | 17 | 106 | 20 |
| | | 17 | 18 | 19 | 20 | 21 | 22 | 23 | 24 | 25 | 26 | 27 | 28 | 29 | 30 | 31 | 32 |

**Figure 24.** Summary first precedencies.

It is obvious to identify extremes especially in developed countries, Germany, France, Austria, Luxembourg, or the Netherlands. Hungary is the dominant post-socialist country.

Similarly, Figure 25 shows the distribution of the number of long precedencies from each group of factors and by country. Switzerland, Luxembourg, Ireland, and Northern Macedonia are the largest number. The distribution shows the influence of the geographical position, especially in Switzerland and Northern Macedonia, due to the relatively high density of neighboring states.

| | | Austria | Belgium | Bulgaria | Croatia | Cyprus | Czechia | Denmark | Estonia | Finland | France | Germany | Greece | Hungary | Ireland | Italy | Latvia |
|---|---|---|---|---|---|---|---|---|---|---|---|---|---|---|---|---|---|
| ECO | all | 0 | 9 | 9 | 1 | 1 | 1 | 0 | 1 | 2 | 0 | 2 | 6 | 3 | 12 | 4 | 0 |
| ECO | positive | 0 | 0 | 0 | 0 | 1 | 1 | 0 | 1 | 2 | 0 | 0 | 0 | 2 | 6 | 2 | 0 |
| ECO | negative | 0 | 9 | 9 | 1 | 0 | 0 | 0 | 0 | 0 | 0 | 2 | 6 | 1 | 6 | 2 | 0 |
| EDU | all | 1 | 0 | 0 | 1 | 0 | 3 | 3 | 0 | 0 | 0 | 0 | 0 | 0 | 8 | 4 | 0 |
| EDU | positive | 0 | 0 | 0 | 0 | 0 | 0 | 1 | 0 | 0 | 0 | 0 | 0 | 0 | 5 | 0 | 0 |
| EDU | negative | 1 | 0 | 0 | 1 | 0 | 3 | 2 | 0 | 0 | 0 | 0 | 0 | 0 | 3 | 4 | 0 |
| EM | all | 0 | 0 | 0 | 0 | 0 | 0 | 0 | 0 | 0 | 0 | 0 | 0 | 0 | 0 | 0 | 0 |
| EM | positive | 0 | 0 | 0 | 0 | 0 | 0 | 0 | 0 | 0 | 0 | 0 | 0 | 0 | 0 | 0 | 0 |
| EM | negative | 0 | 0 | 0 | 0 | 0 | 0 | 0 | 0 | 0 | 0 | 0 | 0 | 0 | 0 | 0 | 0 |
| RD | positive | 0 | 3 | 0 | 0 | 1 | 0 | 5 | 0 | 0 | 0 | 3 | 0 | 0 | 17 | 0 | 0 |
| SOC | negative | 0 | 0 | 0 | 0 | 0 | 13 | 0 | 0 | 0 | 3 | 0 | 0 | 0 | 0 | 0 | 0 |

| | | Lithuania | Luxembourg | Malta | Netherlands | North Macedonia | Norway | Poland | Portugal | Romania | Serbia | Slovakia | Slovenia | Spain | Sweden | Switzerland | United Kingdom |
|---|---|---|---|---|---|---|---|---|---|---|---|---|---|---|---|---|---|
| ECO | all | 7 | 17 | 0 | 9 | 17 | 0 | 1 | 0 | 27 | 0 | 3 | 1 | 1 | 0 | 12 | 15 |
| ECO | positive | 0 | 2 | 0 | 5 | 0 | 0 | 0 | 0 | 0 | 0 | 0 | 1 | 1 | 0 | 9 | 7 |
| ECO | negative | 7 | 15 | 0 | 4 | 17 | 0 | 1 | 0 | 27 | 0 | 3 | 0 | 0 | 0 | 3 | 8 |
| EDU | all | 3 | 9 | 0 | 0 | 3 | 3 | 6 | 0 | 1 | 0 | 2 | 4 | 0 | 1 | 11 | 0 |
| EDU | positive | 0 | 3 | 0 | 0 | 0 | 3 | 0 | 0 | 0 | 0 | 0 | 0 | 0 | 1 | 9 | 0 |
| EDU | negative | 3 | 6 | 0 | 0 | 3 | 0 | 6 | 0 | 1 | 0 | 2 | 4 | 0 | 0 | 2 | 0 |
| EM | all | 0 | 0 | 0 | 1 | 9 | 2 | 0 | 0 | 0 | 0 | 0 | 0 | 0 | 3 | 4 | 0 |
| EM | positive | 0 | 0 | 0 | 1 | 0 | 2 | 0 | 0 | 0 | 0 | 0 | 0 | 0 | 3 | 4 | 0 |
| EM | negative | 0 | 0 | 0 | 0 | 9 | 0 | 0 | 0 | 0 | 0 | 0 | 0 | 0 | 0 | 0 | 0 |
| RD | positive | 0 | 44 | 0 | 0 | 0 | 0 | 0 | 0 | 0 | 0 | 0 | 2 | 0 | 0 | 0 | 3 |
| SOC | negative | 0 | 1 | 0 | 5 | 0 | 0 | 0 | 0 | 0 | 0 | 0 | 0 | 0 | 0 | 12 | 0 |

**Figure 25.** Summary of long precedencies.

*3.4. Compared Precedence and Real Values*

After the precedence analysis, the point order obtained by precedence for individual states was compared with the order determined at fair values. Due to uneven representation of factors in groups of factors, the comparison was made on real and proportionally adjusted data. The data was modified as follows: ProporDate = AllDate/SelectionDate.

Proportional data reflect the weight of the relevant set of factors (ECO, EM, RD, EDU, SOC). In essence, they indicate how many sub-factors are involved in a given group. In large numbers, the group's strength is weakened due to the proportionate approach to the whole. This approach is chosen because the individual factors in the group determine the order that assigns points to each country. In the case of the sum of these points, the sum is greater with a number of factors, which must

be regulated. If the individual factors were equally represented in the groups, the proportional values would be the inverse of the total values (only the national order would be reversed).

Real values, which are recalculated using point order, show a more even distribution than precedence, as can be seen from the percentage distribution (orange line) in Parete charts.

Figures 26–30 shows the order of states in Parete charts. If we compare real and recalculated values with precedents, we find groups of states with similar characteristics.

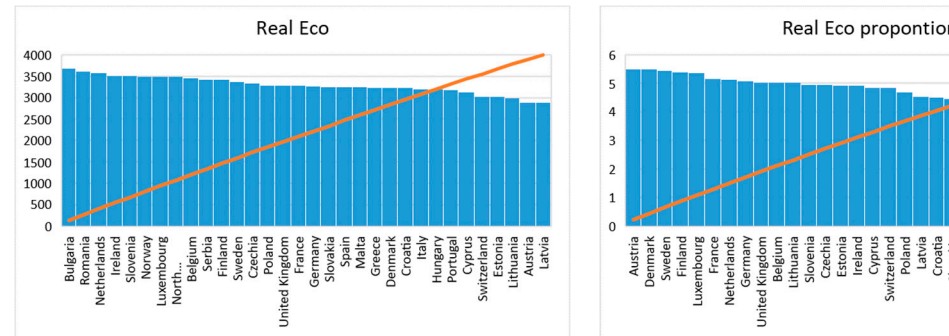

**Figure 26.** Compared real values, Eco factors.

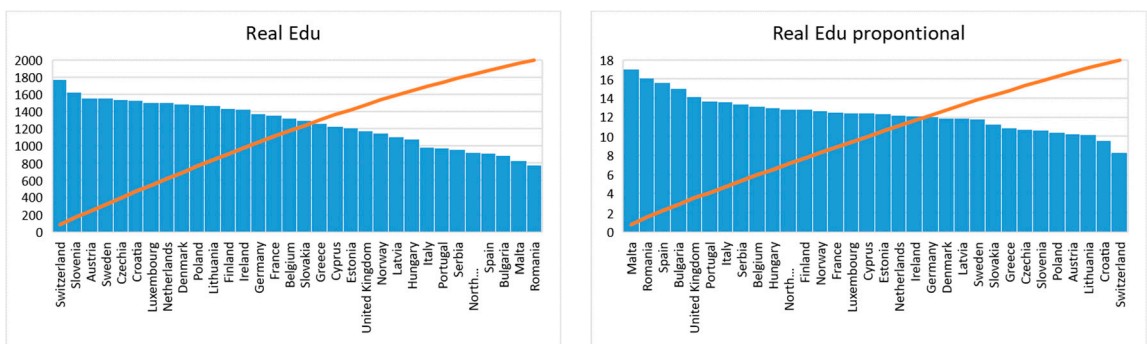

**Figure 27.** Compared real values, Edu factors.

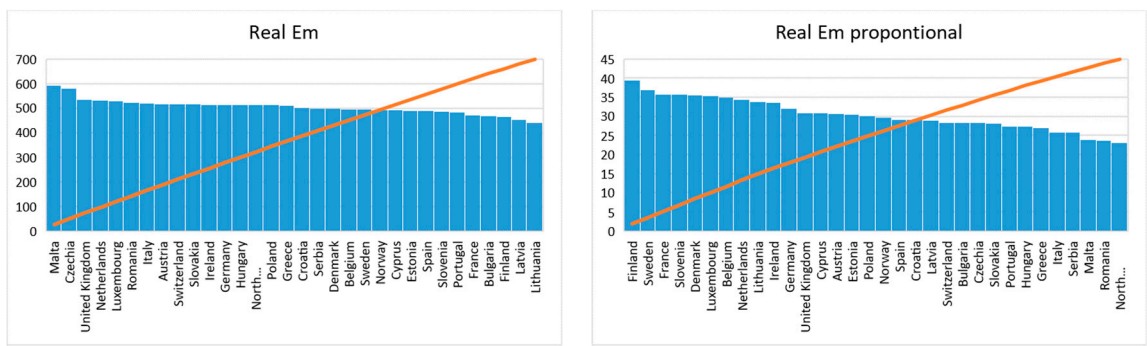

**Figure 28.** Compared real values, EM factors.

There is a group of countries that show high scores for real or recalculated values. Malta, for example, has the highest values in the Em group and the Edu proportional group, but in the precedence analysis it reaches zero values (see Figures 9, 10, 12, 13, 15, 16, 18 and 19, etc.). By comparing individual sub-graphs, it is possible to find out that long precedencies correspond more closely with real values, the first precedence correspond more closely with proportional values.

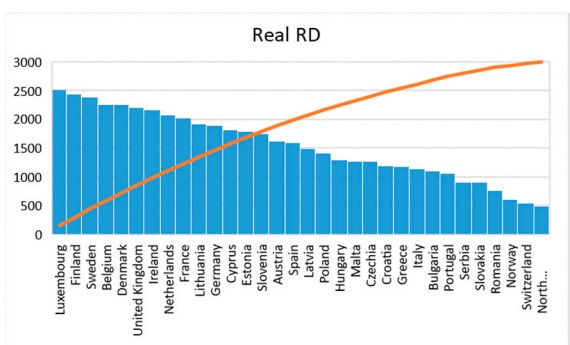 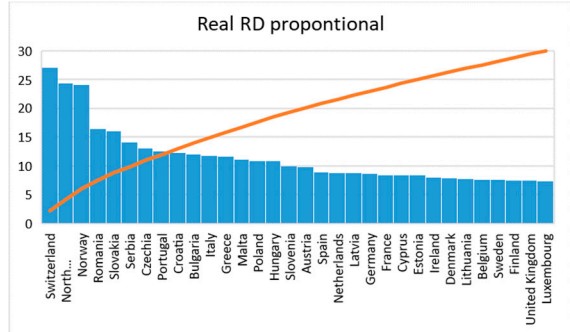

**Figure 29.** Compared real values, RD factors.

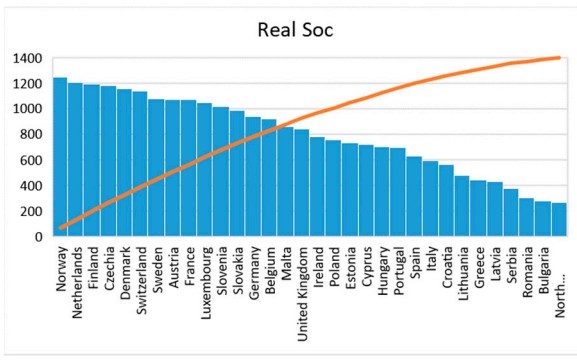 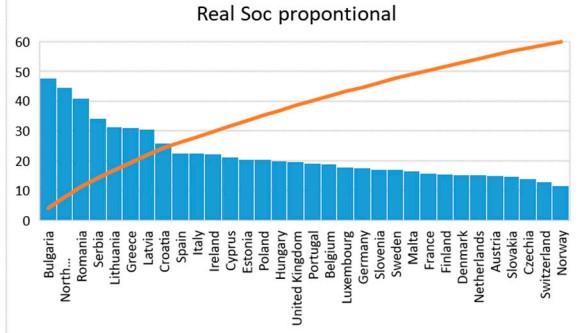

**Figure 30.** Compared real values, Soc factors.

In the Eco group, it can be noted that countries like Romania, Luxembourg, or North Macedonia are similarly rated in long precedents and real values. In contrast, Germany, France, Austria, or Luxembourg are identifiable in early precedents and proportional values.

In the Edu group it is similarly possible to trace connections between Switzerland, Luxembourg, or Poland (real, long precedent), but there is no connection between the first precedents and the proportional values.

If we compare other groups (Em, RD, Soc) we find that the possible dependencies are rather given by groups of states with similar values (for example, in real RD states between UK and Greece), when it is easy to confuse or purposefully distort the order.

The subject of further investigation is the comparison of measured values.

## 4. Conclusions and Discussion

As noted in the literature review, particularly (Efremov and Vladimirova 2019), point out that globalization is one of the key processes and a major feature of the development of the world economy and significantly reflects fundamental changes in the economic policies of the world's leading powers. The current globalization phase is characterized by the eradication of established economic ties, protectionism, trade and customs wars, and sanctions with the aim of increasing economic instability and a general slowdown in GDP growth rates in competing regions. Opinions of the impasse of the globalization process and the advent of globalization of the world economy are increasingly expressed. The authors decided to confirm or refute this view. At the beginning, the dynamics of the globalization level index of the countries of the world was analyzed, reflecting the degree of integration of the country into the global political, economic, and socio-cultural space. An analysis of the distribution of the benefits of globalization among countries was then carried out. It was found that the largest companies are now approaching their FDI (foreign direct investment) because the digital economy allows them to operate globally and to act in foreign markets with a virtual physical presence. New technologies are leading to changes in the content of international business transactions

and brand-new multinational business models are emerging. The authors conclude that there is an apparent slowdown in globalization processes at present. However, it is of a temporary nature and mainly concerns traditional assets, while the transnational flows of new activities caused by the Fourth Industrial Revolution are substantially strengthened. It can be assumed that there is a new phase of globalization where the driving force is digital technology, which fundamentally changes the ideas and approaches to placing the productive forces in the world, seriously changes the existing value chains, and changes the model and strategy of business development.

Just as there are different opinions in the literature on the positive or negative effects of Industry 4.0 or on the degree of impact on employment and quality of life, it is not possible to accurately estimate the impact and significance of the factors examined. Employment in industry statistically registered in GEO/NACE_R2 as M N—Professional, scientific and technical activities; administrative and support service activities should show a significant level of threat to Industry 4.0. However, this group includes not only administrative (i.e., endangered) groups, but also scientific and professional groups, which in turn would have a positive effect in Industry 4.0. It is also not possible to reliably identify the importance of the proportion of women from statistical data; it is a known fact that women are less interested in ICT. From this, it could be concluded that an increase in the proportion of women in certain sectors will lead to greater threats and less adaptability to the consequences of Industry 4.0. However, if, for example, there is an increase in women in a negative identifier related to whether it is a quantitative increase or a share increase. It is obvious to the authors that the enumeration of factors is not complete at this time and the views on the use of individual factors may vary, long-term research, and specific measurable impacts of Industry 4.0 will demonstrate the parameters suitability and their true weight in the future The paper presented a method that has the potential given by simplifying the analysis into a binary form of data and allowing more intensive work with larger data volumes.

During the research, it can be seen that the density of the generated network can significantly affect the resulting number of precedents and results can be distorted. This is evident, for example, in the total number of first precedents in the UK (e.g., Figure 4). However, this distortion is natural because the precedence method takes into account the real geospatial context and the results show the relative isolation of the UK. However, this spatial dependence can also lead to misinterpretation, as demonstrated by the example of Malta, which has no precedence and always includes the last positions in a precedent comparison. However, it is at the top of some groups in comparison to fair values. If we analyzed precedence only, we would not record high real values and misinterpret that they do not exist. However, if we analyze the values with respect to the spatial distribution, then the conclusions of the precedence analysis will be correct, because the relative isolation of Malta overrides the significance of the high value of the relevant factor. In practice, it has again been shown that, in any analysis process, it is necessary to apply a systemic approach and it is not possible to generalize individual results using isolated methods.

The results may be distorted by the fact that the availability and weight of factors are not the same across countries, as reported by (Birkel et al. 2019), German companies are chosen more often for studies and analyses due to the importance of Germany as an industrial nation, as well as the acquired knowledge with Industry 4.0-related technologies, which is already available.

Some authors (Kagermann et al. 2013) presuppose Germany's diminishing role in the Industry 4.0 process because:

> "Germany is the world's leading manufacturing equipment supplier, Germany is uniquely well placed to tap into the potential of this new form of industrialisation. Germany's global market leaders include numerous 'hidden champions' who provide specialised solutions—22 of Germany's top 100 small and medium-sized enterprises (SMEs) are machinery and plant manufacturers, with three of them featuring in the top ten. Indeed, many leading figures in the machinery and plant manufacturing industry consider their main competitors to be domestic ones. Machinery and plant also rank as one of Germany's main exports alongside cars and chemicals. Moreover, German machinery and plant manufacturers expect to maintain their

leadership position in the future. 60% of them believe that their technological competitive advantage will increase over the next five years, while just under 40% hope to maintain their current position."

However, the conclusions of the analyses point to almost similar positions, especially for France, the Netherlands, and Austria. Furthermore, when comparing factors in the border years of the monitored intervals, Austria's dominance in the first precedents is apparent, which means that it has higher values of factors in the spatial context than neighboring states (Figure 9). As Figure 11 shows, Germany has the most precedents in economic factors, especially with a negative impact. Austria is dominant in educational, science, and research-related factors and in social factors. A comparison of short precedents shows discrepancies between developed countries, for example Germany or Austria have a relatively high number of first precedents (Figure 5). By contrast, the UK has a low number of these precedents. It is not possible to determine unequivocally whether this is due to geographical location. According to (Reischauer 2018), a different type of economy may be a possible result. While Germany (and for example Japan too) are prototypical examples for coordinated market economy, the UK (and for example USA) are paradigm examples for a liberal market economy. These different economy types may also shape the basic features of discourses on digital technologies in manufacturing industries.

Another example of spatial dependence is Italy. The linear geographic shape of Italy initiates a high number of links, leading to the assumption of a high number of first precedents. Yet, as shown in Figures 9, 12, 15, 19, 21 and 23, the number of Italy's first precedents is below average, in some cases lower than in the post-communist countries (Czechia, Slovakia, Hungary, and the like). This is due to low readiness for Industry 4.0.

Although manufacturing industry units developing the Industry 4.0 concept in 2018 in Italy achieves overall 17% (Büchi et al. 2020), a relationship between Italy manufacturing companies opened to Industry 4.0 and performance, measured in terms of the application of at least one pillar of 4.0-enabling technologies shows that the average Italian region achieves results 8%, average Europa is 15% and Germany's national average is 25%. The exception are EM negative factors precedencies, where Italy has the highest number of first precedents (Figure 18).

It can be stated that the relationship between the comparison of fair values and precedence has not been proven. In connection with the analysis of individual factors, it can be stated that more developed countries have better positions for positive factors, but may be endangered by negative impacts, less developed ones have the potential of low risk of negative factors.

However, on the basis of these findings, the hypothesis of dependence of individual factors cannot be rejected or confirmed. Although precedence analysis has identified countries in which there are similar numbers of precedencies (both positively and negatively), the significance of the chosen generated infrastructure and the consequent connection with the comparison of real values has not yet been sufficiently investigated. It is clear that the precedence analysis takes the spatial context into account.

The study brings new, structural relationships between the analyzed factors. It shows the geographical distribution of local extremes and indicates global extremes. For regional analyzes it brings new tools and forms of analysis that can be subsequently used for multiagent simulations. Population development and migration due to changes in the values of individual factors can be predicted by the movement of autonomous systems, as part of the analysis is the generation of infrastructure that can serve for the movement of autonomous agents based on a simple rule. Long precedence serves as a decision criterion.

At present, the research does not aim to analyze all available factors, the selection of a group of factors may be questioned, based on other research and recommendations the data will be adjusted. Factors were selected to include a broad spectrum and selected monitored data were to verifiable and traceable.

Further research will focus on three basic areas:

1. Industry 4.0 technology implications. In this area, the implementation areas of Industry 4.0 tools will be systematically surveyed.
2. Area of identification of indicators and evaluation factors. In this area, the applicability of indicators and their impacts will be analyzed.
3. Precedence analysis of indicators development. In this area, the development of local extremes according to groups of factors will be mapped.

**Author Contributions:** M.B.: theoretical background, research question, literature research, selection of factors, analyzes. J.B.: analyzes, literature research, modeling, conclusions and discussion, visualization of results. All authors have read and agreed to the published version of the manuscript.

**Funding:** This research was funded by Silesian university in Opava grant no. SGS/19/2019, "Application of Customer Relationship Management Systems in Small and Medium-sized Enterprises accepted in 2019", and grant no. SGS 2/2019 "Tourism of the Moravian-Silesian Region in the context of sustainable development" and the APC was funded by the University of Zilina.

**Conflicts of Interest:** The authors declare no conflict of interest.

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
