# Peer review of "Local Extremes of Selected Industry 4.0 Indicators in the European Space—Structure for Autonomous Systems"

_jrfm, doi:10.3390/jrfm13010013_

Round 1
Reviewer 1 Report
Dear authors, thank you for your submission to JRFM. I believe that your results as such are very interesting, but must be embedded better in extant literature. Please find some suggestions below:
1) You use a lot of news sites (CT24), conferences, or from Czech origin. This is fine, but you must complement them with academic, international references (preferably journals) whenever possible. This is especially required for the definition of Industry 4.0 or for showing the extant state of research on Industry 4.0. Further, some statements in the introduction are not referenced at all. Some statements could be referenced by including references for economic/social factors (triple bottom line of sustainability) in the light of Industry 4.0. Since there are too few academic references on Industry 4.0 for an academic article so far, please find some suggestions below:
Beier, G., Niehoff, S., Ziems, T., & Xue, B. (2017). Sustainability aspects of a digitalized industry–A comparative study from China and Germany. International Journal of Precision Engineering and Manufacturing-Green Technology, 4(2), 227-234.
Beier, G., Niehoff, S., & Xue, B. (2018). More Sustainability in Industry through Industrial Internet of Things?. Applied Sciences, 8(2), 219.Birkel, H. S., Veile, J. W., Müller, J. M., Hartmann, E., & Voigt, K. I. (2019). Development of a risk framework for Industry 4.0 in the context of sustainability for established manufacturers. Sustainability, 11(2), 384.
Gnezdova, J. V., Rudakova, E. N., & Zvyagintseva, O. P. (2019). Systemic Contradictions of Modern Economic Systems That Hinder Formation and Development of Industry 4.0 in the Conditions of Knowledge Economy’s Formation and Methods of Overcoming Them. In Industry 4.0: Industrial Revolution of the 21st Century (pp. 211-218). Springer, Cham.
Horváth, D., & Szabó, R. Z. (2019). Driving forces and barriers of Industry 4.0: Do multinational and small and medium-sized companies have equal opportunities?. Technological Forecasting and Social Change, 146, 119-132.Liao, Y., Deschamps, F., Loures, E. D. F. R., & Ramos, L. F. P. (2017). Past, present and future of Industry 4.0-a systematic literature review and research agenda proposal. International Journal of Production Research, 55(12), 3609-3629.
Maresova, P., Soukal, I., Svobodova, L., Hedvicakova, M., Javanmardi, E., Selamat, A., & Krejcar, O. (2018). Consequences of Industry 4.0 in business and economics. Economies, 6(3), 46.Müller, J. M., & Voigt, K. I. (2018). Sustainable industrial value creation in SMEs: A comparison between industry 4.0 and made in China 2025. International Journal of Precision Engineering and Manufacturing-Green Technology, 5(5), 659-670.
Müller, J. M. (2019). Antecedents to Digital Platform Usage in Industry 4.0 by Established Manufacturers. Sustainability, 11(4), 1121.
Ragulina, Y. V., Shkodinsky, S. V., Mishchenko, V. V., & Romanova, Y. A. (2019). Scenarios of Development of Industry 4.0 in the Conditions of Knowledge Economy’s Formation and Their Consequences for Modern Economic Systems. In Industry 4.0: Industrial Revolution of the 21st Century (pp. 227-234). Springer, Cham.2) There is no discussion with extant literature so far, please discuss how your findings contribute to extant knowledge. Further, please extend limitations and suggestions for future research in your study.
Author Response
Dear colleague
Thank you for your suggestive reviews. I worked up some of them. We tried to spread the quoted sources evenly. There are certainly many sources, but I do not consider it serious to mention in the article sources that I have not studied. We have added one resource specifying Risk factors related to the behavioral behavior in Industry 4.0 process. For the sake of clarity and at the recommendation of the other opponent, we have added a table summarizing the priority citation sources. The table shows the distribution of resources, 9 resources from WoS conferences, 10 resources from Q1 / Q2 magazines, 2 resources from Q2 / Q3 magazines and 1 resource from WoS magazine without the quoted quartile. 1 professional periodical, 1 professional internet periodical, 1 scientific monographic. Of these, 14 are focused on Industry 4.0, including a summary of knowledge and terminology. These sources were supplemented with up-to-date information from information sources (television). In the discussion, we added the findings and benefits of the study to your recommendation: The study brings new, structural relationships between the analyzed factors. It shows the geographical distribution of local extremes and indicates global extremes. For regional analyzes it brings new tools and forms of analysis that can be used for multiagent simulations. Population development and migration due to changes in individual factors can be predicted by the movement of autonomous systems, as part of the analysis is the generation of infrastructure that can serve for the movement of autonomous agents based on a simple rule. Long precedence serves as a decision criterion.
Thank you for your suggestions and we look forward to further cooperation.
Reviewer 2 Report
For the easy to read and understand literature, authors can use a table to list previous studies, and to point out: What pros and cons of those studies? what research gaps were? Why need this new study? Authors use mutliagent analysis as research method, so authors need to introduce what is multiagent analysis? How to use it to analysis? Based on what kind of literature, the 29 factors related to Industry 4 were selected? authors should describe it. In Analysis using Precedence section. why the analysis use first and long precedencies analysis? what it mean? The results of analysis, authors should use a table to show what have been analyzed and what results were. In discussion section, authors should enhance discussion and connect with previous studies. What contribution or new to academic of this study? What contribution to practice or management of this study? Authors should add it. Line 786, what mean FDI is?Author Response
Thank you for your suggestions - 1 citation of Q1-Q2 magazine was added, other citations were not added because the authors did not study them. We have 12 citations to WoS Journals Q1-Q3, which we consider to be sufficient. We have added a table the citations specification as requested by the other reviewer.- the concept of multiagent analysis was clarified
- the first and long precedents and their use in research have been described
- results tables added
- the benefits of the study were added
- was specified term FDI
- language correction was performed
Round 2
Reviewer 1 Report
Dear authors, thank you for your revisions. Unfortunately, I must say that the main weakness of the paper has not been addressed: It is not sufficient to base large parts of the introduction on news reports and non-scientific sources (I am not talking about the actual analysis made, but about the introduction and emdeddedness in extant literature, which is not academic standard). The answer "There are certainly many sources, but I do not consider it serious to mention in the article sources that I have not studied" puzzles me. There is the need to base your introduction on academic references, and if you haven't studied such articles yet, you must do it now. Otherwise, your article is not suitable for international academic standards. It is neccessary to base your statements about what Industry 4.0 is on actual scientific references, among other statements. Please find the recommendations of the first review below, which has not been solved so far:
1) You use a lot of news sites (CT24), conferences, or from Czech origin. This is fine, but you must complement them with academic, international references (preferably journals) whenever possible. This is especially required for the definition of Industry 4.0 or for showing the extant state of research on Industry 4.0. Further, some statements in the introduction are not referenced at all. Some statements could be referenced by including references for economic/social factors (triple bottom line of sustainability) in the light of Industry 4.0. Since there are too few academic references on Industry 4.0 for an academic article so far, please find some suggestions below:
Beier, G., Niehoff, S., Ziems, T., & Xue, B. (2017). Sustainability aspects of a digitalized industry–A comparative study from China and Germany. International Journal of Precision Engineering and Manufacturing-Green Technology, 4(2), 227-234.
Beier, G., Niehoff, S., & Xue, B. (2018). More Sustainability in Industry through Industrial Internet of Things?. Applied Sciences, 8(2), 219.
Birkel, H. S., Veile, J. W., Müller, J. M., Hartmann, E., & Voigt, K. I. (2019). Development of a risk framework for Industry 4.0 in the context of sustainability for established manufacturers. Sustainability, 11(2), 384.
Gnezdova, J. V., Rudakova, E. N., & Zvyagintseva, O. P. (2019). Systemic Contradictions of Modern Economic Systems That Hinder Formation and Development of Industry 4.0 in the Conditions of Knowledge Economy’s Formation and Methods of Overcoming Them. In Industry 4.0: Industrial Revolution of the 21st Century (pp. 211-218). Springer, Cham.
Horváth, D., & Szabó, R. Z. (2019). Driving forces and barriers of Industry 4.0: Do multinational and small and medium-sized companies have equal opportunities?. Technological Forecasting and Social Change, 146, 119-132.
Liao, Y., Deschamps, F., Loures, E. D. F. R., & Ramos, L. F. P. (2017). Past, present and future of Industry 4.0-a systematic literature review and research agenda proposal. International Journal of Production Research, 55(12), 3609-3629.
Maresova, P., Soukal, I., Svobodova, L., Hedvicakova, M., Javanmardi, E., Selamat, A., & Krejcar, O. (2018). Consequences of Industry 4.0 in business and economics. Economies, 6(3), 46.
Müller, J. M., & Voigt, K. I. (2018). Sustainable industrial value creation in SMEs: A comparison between industry 4.0 and made in China 2025. International Journal of Precision Engineering and Manufacturing-Green Technology, 5(5), 659-670.
Müller, J. M. (2019). Antecedents to Digital Platform Usage in Industry 4.0 by Established Manufacturers. Sustainability, 11(4), 1121.
Ragulina, Y. V., Shkodinsky, S. V., Mishchenko, V. V., & Romanova, Y. A. (2019). Scenarios of Development of Industry 4.0 in the Conditions of Knowledge Economy’s Formation and Their Consequences for Modern Economic Systems. In Industry 4.0: Industrial Revolution of the 21st Century (pp. 227-234). Springer, Cham.
Author Response
Dear colleague,
we have incorporated comments according to your requirements. We've added 13 links to academic resources (typically featured in WoS Q1 and Q2)
We added a discussion in relation to the Triple Bottom Line.
In particular, we have modified the chapter Theoretical Background and Literature review and current state of knowledge.
We extended the conclusion and added suggestions for future research.
Thank you for your inspirational comments.
best regards, authors
Round 3
Reviewer 1 Report
Dear authors,
thank you for your revisions. I believe that the embeddedness in scientific literature has been improved substantially.